

# Coal fly ash: Linking immersion freezing behavior and physico-chemical particle properties

Sarah Grawe[1], Stefanie Augustin-Bauditz[1,*], Hans-Christian Clemen[2], Martin Ebert[3], Stine Eriksen Hammer[3], Jasmin Lubitz[1], Naama Reicher[4], Yinon Rudich[4], Johannes Schneider[2], Robert Staacke[5], Frank Stratmann[1], André Welti[1,**], and Heike Wex[1]

[1]Leibniz Institute for Tropospheric Research, Experimental Aerosol and Cloud Microphysics Department, Leipzig, Germany
[2]Max Planck Institute for Chemistry, Particle Chemistry Department, Mainz, Germany
[3]Darmstadt University of Technology, Institute of Applied Geosciences, Darmstadt, Germany
[4]Weizmann Institute of Science, Department of Earth and Planetary Sciences, Rehovot, Israel
[5]University of Leipzig, Felix Bloch Institute for Solid State Physics, Division of Nuclear Solid State Physics, Leipzig, Germany
*Now at: Deutscher Wetterdienst, Hamburg, Germany
**Now at: Finnish Meteorological Institute, Helsinki, Finland

**Correspondence:** Sarah Grawe (grawe@tropos.de)

**Abstract.**

To date, only a few studies have investigated the potential of coal fly ash particles to trigger heterogeneous ice nucleation in cloud droplets. The presented measurements aim at expanding the sparse dataset and improving process understanding of how physico-chemical particle properties influence the freezing behavior of coal fly ash particles immersed in water.

5  Firstly, immersion freezing measurements were performed with two single particle techniques, i.e., the Leipzig Aerosol Cloud Interaction Simulator and the Spectrometer for Ice Nuclei. The effect of suspension time on the efficiency of the coal fly ash particles when immersed in a cloud droplet is analyzed based on the different residence times of the two instruments and employing both dry and wet particle generation. Secondly, two cold stage setups, one using microliter sized droplets (Leipzig Ice Nucleation Array) and one using nanoliter sized droplets (Weizman Supercooled Droplets Observation on Microarray setup) were applied.

10  We found that coal fly ash particles are comparable to mineral dust in their immersion freezing behavior when being dry-generated. However, a significant decrease in immersion freezing efficiency was observed during experiments with wet-generated particles in LACIS and SPIN. The efficiency of wet-generated particles is in agreement with the cold stage measurements. In order to understand the reason behind the deactivation, a series of chemical composition, morphology, and crystallography analyses (single particle mass spectrometry, scanning electron microscopy coupled with energy dispersive X-ray microanalysis, X-ray diffraction analysis) was performed with dry- and wet-generated particles. From these investigations, we conclude that anhydrous $CaSO_4$ and $CaO$, which, if investigated in pure form, show the same qualitative immersion freezing behavior as observed for dry-generated coal fly ash particles, contribute to triggering heterogeneous ice nucleation at the particle-water interface. The observed deactivation in contact with water is related to changes of the particle surface properties





which are potentially caused by hydration of $CaSO_4$ and $CaO$. The contribution of coal fly ash to the ambient population of ice nucleating particles therefore depends on whether and for how long particles are immersed in cloud droplets.

## 1 Introduction

It is known that naturally occurring aerosol such as biological particles (e.g., bacteria, pollen, spores) and mineral dust are acting as Ice Nucleating Particles (INPs; Hoose and Möhler, 2012 and references therein). In contrast, there is ongoing discussion about the impact of anthropogenic aerosol emissions on the concentration of atmospheric INPs (Szyrmer and Zawadzki, 1997). The strongest source of anthropogenic aerosol is the combustion of fossil fuels, where primary particles such as carbonaceous aerosol and ash, as well as secondary particles from gaseous precursors are generated.

Carbonaceous aerosol, such as soot, which is a product of incomplete combustion of organic material, has been shown to be ice nucleation active (e.g., DeMott, 1990; Diehl and Mitra, 1998; Fornea et al., 2009). However, there are large discrepancies between studies investigating the ice nucleation ability of soot, which might be related to source and/or mixing state of the particles (Kanji et al., 2017). Hoose and Möhler (2012) summarize that "soot is a generally worse ice nucleus than mineral dust". In contrast to soot and other carbonaceous aerosol types, ash contains only a limited amount of carbon. Defined as the solid remains from the combustion of organic substances, e.g., wood or fossil fuels, it consists mostly of the non-combustible constituents in the fuel, i.e., mineral inclusions and atoms other than C and H, e.g., K, Ca, Mn, Fe, etc. (Flagan and Seinfeld, 1988a). A distinction is made between the fine ash fraction, i.e., fly ash, that is emitted during combustion together with flue gases, and the coarse ash fraction, i.e., bottom ash, that remains in the fireplace.

Coal is difficult to substitute in the energy mix of most industrial countries and hence only slowly replaced by renewable energy sources (U. S. Energy Information Administration, 2017). In total, 6711 coal-fired power plants (units 30 MW and larger; endcoal.org, status: July 2017) are in operation worldwide, producing 600 Mt/a of coal ash (Ahmaruzzaman, 2010). The vast majority of this mass is not emitted into the atmosphere, as coal-fired power plants are equipped with different types of particle removal technology to clean flue gases from Coal Fly Ash (CFA). Estimating CFA emissions is not trivial, because filtering systems show varying efficiencies and part of the collected CFA is emitted during disposal (Mueller et al., 2013). A rough assessment was given by Smil (2008), estimating that 30 Mt/a of CFA are released into the atmosphere worldwide. Reff et al. (2009) state that coal combustion causes $PM_{2.5}$ emissions of ~0.5 Mt/a in the USA. In addition to a large uncertainty of these estimates, there is no detailed information about temporal and spatial variability of CFA emission and dispersion which is important for assessing the effect of CFA particles on cloud formation and glaciation.

A lot of research has been conducted already in the field of CFA sample characterization for identifying CFA particles in the atmosphere. This was mainly driven by concerns about the negative effects of CFA particles on human health (e.g., Davison et al., 1974; Damle et al., 1982; Yi et al., 2006; and references therein). These studies show that CFA has a complex and highly variable composition. Except for some trace elements whose contents are heterogeneously distributed among different size fractions, CFA composition is comparable to mineral dust, making it difficult to identify via single particle mass spectrometry (Cziczo et al., 2004, 2006; Kamphus et al., 2010). CFA particles are, in contrast to irregularly shaped mineral dust particles,



often perfectly spherical because of their generation process, where minerals melt and form spherical droplets that retain their shape upon solidification (Damle et al., 1982; Flagan and Seinfeld, 1988a). However, shape is not a perfect criterion for identifying CFA, as other high temperature processes such as fuel oil combustion or metal processing, also emit spherical fly ash particles. In addition, there are various aerosol types which occur in spherical shapes, e.g., biological particles (Huffman et al., 2012), tar balls (Laskin et al., 2006), or deliquesced salt particles (Freney et al., 2009). In conclusion, a reliable identification of CFA particles is not trivial and probably requires a combination of chemical composition and morphology analysis.

Concerning the ice nucleation activity of ash particles, only few studies have been published so far (Havlíček et al., 1993; Umo et al., 2015; Garimella, 2016; Grawe et al., 2016). Havlíček et al. (1993) investigated chemical composition and ice nucleation characteristics of CFA from 9 different power plants in former Czechoslovakia focusing on the effect of water-soluble material in the samples. The chemical composition analysis showed that the water-soluble fraction of the samples varied between 0.43 and 1.34 wt% and mainly consisted of anhydrite (anhydrous $CaSO_4$). Ice nucleation experiments were carried out with two methods. Firstly, polydisperse CFA particles were aerosolized in a thermodiffusion chamber subsaturated with respect to liquid water at -15°C, i.e., only deposition nucleation was investigated. Secondly, suspensions of CFA in distilled water were used to produce droplets onto a cooled plate (cold stage), i.e., immersion freezing was investigated. The ice nucleation efficiency of the original CFA material, samples freed from water-soluble components, and the water-soluble components alone was quantified. Immersion freezing was found to be less efficient than deposition nucleation for all of the samples. Also, samples freed from water-soluble components were up to three orders of magnitude less efficient in the deposition mode than the untreated samples. However, when the water-soluble components alone were investigated, they showed surprisingly low efficiency, i.e., the water-soluble components increased the ice nucleation efficiency only when associated with the CFA particles. This finding illustrates the complex interplay of physico-chemical particle properties and freezing behavior.

Four ash samples including CFA, coal bottom ash, wood bottom ash, and bottom ash from a domestic oven were investigated by Umo et al. (2015). The immersion freezing behavior was quantified using a cold stage setup (Whale et al., 2015). In comparison to the bottom ashes, CFA was more efficient at nucleating ice between -17 and -27°C, showing a strong increase starting at -16°C and an apparent plateau below roughly -24°C. The bottom ashes behaved similar to one another, with a slight trend of coal bottom ash being less efficient and wood bottom ash being more efficient.

Garimella (2016) investigated four different CFA samples from the U.S. concerning their deposition nucleation and immersion freezing behavior using the SPectrometer for Ice Nuclei (SPIN; Droplet Measurement Technologies, Inc.). In this study, particles were dry-generated and size-selected. Activated fractions of 1 % were observed at $T < -30$°C for deposition nucleation and at $T < -20$°C for immersion freezing, which is contradictory to what Havlíček et al. (1993) reported. When comparing immersion freezing measurements by Garimella (2016) and Umo et al. (2015), a discrepancy of more than one order of magnitude was found, with the cold stage measurements being below the immersion freezing measurements with SPIN. In addition, Garimella (2016) showed that 300 nm particles are more efficient per unit surface area than 700 nm particles, possibly indicating that trace metals, which are enriched in smaller particles due to size dependent cooling rates, could contribute to the immersion freezing efficiency. This could explain why the results by Umo et al. (2015), where the size distribution of immersed particles had a mode diameter of ~10 $\mu$m, were much lower.



In a previous study (Grawe et al., 2016), we investigated the freezing behavior of different ash samples, including bottom ash from wood and brown coal burning, as well as CFA. Experiments were performed with the Leipzig Aerosol Cloud Interaction Simulator (LACIS; Hartmann et al., 2011), a laminar flow tube in which single, size-selected particles are activated to droplets and cooled down to investigate immersion freezing (see Sec. 2.3.1). It was found that dry-generated CFA particles

showed the highest immersion freezing efficiency of the examined samples, being only slightly less efficient below -27 °C than a previously investigated K-feldspar sample (Augustin-Bauditz et al., 2014). Interestingly, a change in immersion freezing efficiency could be seen in transition to wet particle generation, i.e., producing ash suspensions which were sprayed with an atomizer and sent through a dryer. In this case, a decrease towards the limit of detection was observed. As the size of dry- and wet-generated particles was identical, the deactivation contradicts the proposed hypothesis of Garimella (2016) that the size

dependent enrichment of trace elements causes the discrepancy between measurements with single particle instruments and cold stages.

The presented study intends to function as a follow-up to our previous paper and aims at answering the following questions:

– Do CFA samples from different power plants feature a similar immersion freezing behavior?

– Is the deactivation in transition from dry to wet particle generation observable for different CFA samples?

– Is it possible to find a connection between physico-chemical sample properties and the observed immersion freezing behavior?

– Which particle generation technique (dry or wet particle generation) or measurement method (single particle vs. cold stage) is appropriate for representing atmospheric processes after CFA emission?

To answer these questions, four CFA samples from German power plants were investigated as immersion freezing INPs.

Additional sample characterization with respect to chemical composition, morphology, and crystallography, was performed and used for interpretation of the immersion freezing results.

## 2   Materials and methods

The immersion freezing and particle characterization measurements of CFA particles were performed during a campaign at TROPOS in November 2016 together with collaborators from the Ice Nuclei research UnIT (INUIT). The main setup (see Fig.

1) consisted of particle generation, size selection, and distribution of the size selected aerosol to the following instruments: 1) LACIS, 2) SPIN, 3) the Aircraft-based Laser ABlation Aerosol MAss spectrometer (ALABAMA), and 4) the multi Micro INertial Impactor (MINI) sampling particles onto substrates for Environmental Scanning Electron Microscopy coupled with Energy Dispersive X-ray spectroscopy (ESEM/EDX). In addition to LACIS and SPIN, immersion freezing measurements were performed with two cold stage setups: 1) the Leipzig Ice Nucleation Array (LINA), and 2) the WeIzmann Supercooled Droplets

Observation on Microarray (WISDOM) setup. For this, suspensions of CFA in water were prepared using the bulk material, which is why further bulk analyses regarding chemical composition and crystallography were performed.





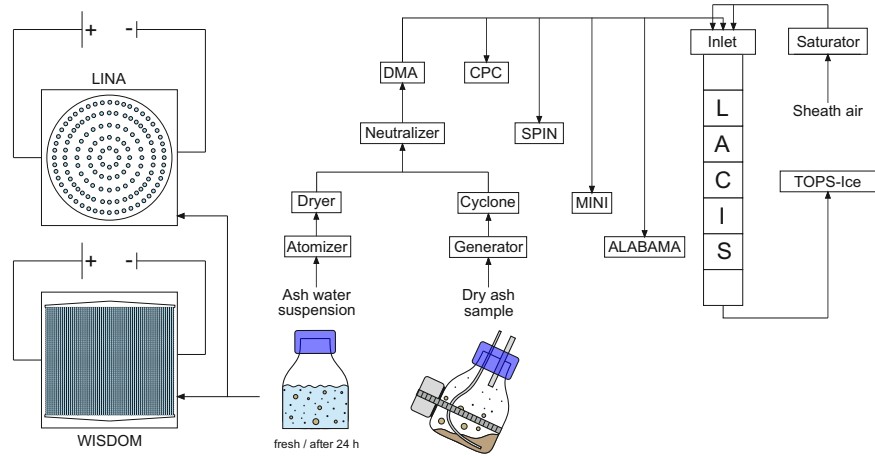

**Figure 1.** Instrumental setup during the INUIT campaign.

## 2.1 Origin of the samples

The CFA samples were taken from the electrostatic precipitators[1] of four coal-fired power plants in Germany. It is neither known which flue gas desulfurization technique is applied in the power plants, nor whether the electrostatic precipitators are installed up- or downstream of the flue gas desulfurization systems. CFA1 is identical to the CFA sample from Grawe et al.

5  (2016), i.e., from the Lippendorf power plant situated 15 km south of Leipzig, Germany. The other power plants shall remain anonymous. CFA1, CFA2, and CFA4 are from brown (sub-bituminous) coal combustion, CFA3 is from black (bituminous) coal combustion.

Lime (CaO), anhydrite ($CaSO_4$), and gypsum ($CaSO_4 \cdot 2H_2O$) from Merck KGaA (Darmstadt, Germany) were used for additional investigations.

## 2.2 Sample preparation and particle generation

Two different kinds of particle generation were used in connection with the LACIS and SPIN immersion freezing experiments: dry particle generation, i.e., aerosolization of particles from dry ash powder, and wet particle generation, i.e., atomization of a CFA-water suspension. Suspensions were also used for the experiments with LINA and WISDOM.

---

[1]Electrostatic precipitators work on the principle of charging the particles and subsequently sending the flow through an electric field. Particles then migrate to the oppositely charged electrode and hence particulate matter is removed from the flue gas (Flagan and Seinfeld, 1988b). The precipitator itself does not alter particle properties like morphology or chemical composition, only number and mass size distribution are changed (Yi et al., 2006). However, it has been argued that particles which are not captured, potentially contain a larger amount of species condensing from the gas phase onto the CFA surface upon cooling (Parungo et al., 1978).



### 2.2.1 Dry particle generation

The dry CFA samples were placed into an aerosol generator operating via pressurized air and an electric imbalance motor
(Grawe et al., 2016). The samples were not sieved prior to aerosol generation. The aerosol was sent through a mixing bottle
and a cyclone ($D_{50}$ = 500 nm) to reduce the amount of large particles in the flow. Further downstream, a neutralizer was passed,
before a Differential Mobility Analyzer (DMA, Vienna type, medium) was used for size selection. A mobility diameter of 300
nm was chosen for the immersion freezing experiments with LACIS for two reasons. Firstly, electrostatic precipitators have
a minimum collection efficiency for particle sizes between 0.1 and 1 $\mu$m (Flagan and Seinfeld, 1988b; Nóbrega et al., 2004;
Kim et al., 2012), meaning that CFA particles in this size range are more likely to be emitted compared to smaller or larger
particles. Secondly, 300 nm particles will experience relevant atmospheric residence times once emitted (Jaenicke, 1978).

Afterwards, the quasi-monodisperse aerosol was distributed to a Condensation Particle Counter (CPC), LACIS, SPIN, MINI,
and ALABAMA. ALABAMA measurements of vacuum aerodynamic diameter were used for multiply charged fraction deter-
mination. The multiple charge correction was performed on frozen fraction ($f_{ice}$) values. The method and results are described
in Sec. S7. Unfortunately, only data acquired in parallel to ALABAMA measurements could be corrected. Data without multi-
ple charge correction are indicated.

### 2.2.2 Preparation of CFA suspensions for cold stages and wet particle generation

The CFA-water suspensions for LACIS, SPIN, and LINA measurements were prepared following the description in Umo et al.
(2015). Briefly, a certain amount of CFA (LINA: 0.1 wt%, LACIS and SPIN: 0.5 wt%) was mixed with distilled water and
ultrasonicated (RK100H Sonorex Super, BANDELIN electronic GmbH & Co. KG) for 10 min. Afterwards, the suspension was
stirred with a magnetic stirrer for 24 h. This approach was chosen to allow comparability to results by Umo et al. (2015) and
Grawe et al. (2016). The procedure helps breaking up large aggregates and hence prevents fast sedimentation that would lead to
an uneven distribution of material in the droplets for LINA. As sedimentation is no limiting factor for wet particle generation (a
flask shaker was used), LACIS measurements were performed with both, the standard suspensions (ultrasonification and 24 h
stirring) and suspensions that were prepared right before the experiment by simply mixing 0.5 wt% of CFA with distilled water.
In this way, particles were in suspension for not more than 5 min before being used for LACIS measurements. Due to instrument
availability, SPIN measurements could only be performed with the standard suspensions. The suspensions, either fresh or
standard, were sprayed with an atomizer (similar to TSI Model 3076) and resulting droplets were sent through a diffusion
dryer. Then, size selection of the particles by the DMA and distribution to LACIS, SPIN, CPC, MINI, and ALABAMA took
place.

In contrast to LINA measurements, size selection of the CFA samples was necessary for WISDOM because large particles
that are present in the original sample would clog the microfluidic device which is used for droplet production (see Sec. 2.3.4).
Size selection took place by running dry particle generation (aerosol generator, mixing bottle, cyclone) for several hours and
collecting the accumulated material from the cyclone ($D_{50}$ = 450 nm). During this procedure, coarse material was deposited in
the mixing bottle and a sub-fraction of the bulk, hereafter referred to as fine CFA, remained in the cyclone. Suspensions of 0.1



wt% fine CFA in distilled water were mixed for 3 cycles of 30 s each with 10 s break in a small volumes sonicator (Hielscher UP200St) and were used for droplet production and immersion freezing experiments within 2 min.

## 2.3 Immersion freezing instrumentation

### 2.3.1 LACIS

5 LACIS is a laminar flow tube consisting of seven 1 m long sections, each temperature controlled by individual thermostats. At the inlet, the aerosol flow is enclosed by a humidified sheath flow. As a result, a stable 2 mm wide particle beam is created along the LACIS centerline, ensuring that all particles experience identical thermodynamic conditions. Supersaturation is created by adjusting the dew point of the sheath air and the wall temperature. Like this, each particle is activated to a droplet before being cooled down for immersion freezing investigation. The ice nucleation time in LACIS is 1.6 s.

10 Supercooled liquid droplets and ice particles coexist at the outlet of the tube in a certain temperature range above the homogeneous freezing limit. The Thermo-stabilized Optical Particle Spectrometer for the detection of Ice (TOPS-Ice; Clauß et al., 2013) is used to determine the phase state of the hydrometeors and from this $f_{ice}$. The measurement principle exploits the difference in scattering properties, i.e., depolarization, between non-spherical ice particles and spherical liquid droplets.

At least 2000 hydrometeors were classified for each LACIS data point presented in this study. Occasionally, three or more 15 data points of separate measurements under the same conditions were averaged. In these cases, the $f_{ice}$ error is indicated by the standard deviation of the separate measurements. Otherwise, a Poisson error is given depending on the total number of classified hydrometeors in a single measurement. The temperature error of $\pm$ 0.3 K is defined by the temperature stability of the thermostats. The ice nucleation active surface site density $n_s$ was calculated from Eq. 1 assuming the particle surface area $A_p$ to be equal to the surface area of a sphere with a diameter of 300 nm.

$$20 \quad n_s(T) = -\frac{\ln(1 - f_{ice}(T))}{A_p} \qquad (1)$$

### 2.3.2 SPIN

SPIN is a Continuous Flow Diffusion Chamber (CFDC) which has been described in detail by Garimella et al. (2016). In contrast to LACIS, the fraction of particles active as INPs (activated fraction $AF$) is calculated by dividing the number of ice crystals detected with an Optical Particle Counter (OPC) by the total number of aerosol particles measured with a CPC. A 25 threshold size of 3 $\mu$m was used to identify ice crystals in the OPC signals. The temperature uncertainties represent the highest and lowest deviations from the average lamina temperature in the chamber. When compared to LACIS measurements, SPIN data provide information on how immersion freezing results are affected by the different residence times in the two instruments. Ice nucleation times in SPIN depend on thermodynamic conditions and are between 8 and 12 s. In addition to the cyclone, an impactor (TSI, 0.071 cm orifice) with $D_{50}$ = 500 nm was used upstream of the SPIN inlet to reduce the amount of multiply 30 charged particles in the CFA aerosol. Hence, no multiple charge correction was applied to the SPIN data.





### 2.3.3 LINA

Based on the Bielefeld Ice Nucleation ARraY (BINARY; Budke and Koop, 2015), the newly developed Leipzig Ice Nucleation Array (LINA) is a cold stage setup for investigating immersion freezing. 90 suspension droplets, each 1 $\mu$l in volume, are placed into separate compartments onto a hydrophobic glass slide. The compartments, realized by a perforated aluminum plate

covered with a second glass slide, prevent interaction between the droplets, e.g., via the Wegener-Bergeron-Findeisen process or splintering while freezing. Also, the compartments suppress evaporation of the droplets. A cooling stage (Linkam LTS120) with a 40 x 40 mm Peltier element is used for cooling the sample array at a rate of 1 K min$^{-1}$. A thin layer of squalene oil on top of the Peltier element guarantees direct contact to the glass slide and improves heat transfer away from the droplets. The setup is situated in an aluminum housing which is purged with particle free, dry air during the experiment. See Chen et al.

(2018) for details on the temperature calibration routine.

The determination of $f_{\text{ice}}$ is almost fully automated. A digital camera coupled with an LED dome light takes images every 6 s which is equal to a temperature resolution of 0.1 K at a cooling rate of 1 K min$^{-1}$. Parts of the LED light are shielded with a cardboard ring to cause ring-shaped structures being reflected from the liquid droplets. As the reflective properties of a droplet change upon freezing, the reflection of the ring vanishes directly after the phase change. The images, each relating to

a certain temperature, are later imported into a computer program that detects the number of rings. From this, $f_{\text{ice}}(T)$ can be derived. See Appendix A for details on the correction of LINA data with respect to background INPs, calculation of $n_{\text{s}}$, and error estimation.

### 2.3.4 WISDOM

The WISDOM setup (Reicher et al., 2018) was used to study the immersion freezing of the fine CFA fraction. WISDOM is a

freezing array of monodisperse nanoliter droplets that are produced on a microfluidic device and subsequently arrange into an array of chambers based on Schmitz et al. (2009). Droplets are suspended in an oil mixture, consisting of mineral oil (Sigma Aldrich) stabilized with 2 wt% nonionic surfactant (span80®, Sigma Aldrich). Pure water droplets within the device can be supercooled to below -35°C, where first freezing occurs, i.e., above this temperature no correction regarding background INPs is necessary. Temperature accuracy of WISDOM is 0.34°C. Freezing is observed by a microscope (BX51 Olympus with 10X

objective and transmission mode) and detected for each droplet individually when the optical brightness of the droplet decreases due to the formation of ice crystals. The microfluidic devices are fabricated in the laboratory from polydimethylsiloxane and attached to a 1 mm microscope slide using oxygen plasma treatment. Freezing experiments are conducted in a commercial cryostage (Linkam THMS600) at a cooling rate of 1 K min$^{-1}$.

$n_{\text{s}}$ was determined according to Eq. 2, with the droplet volume $V_{\text{drop}}$ = 478 $\pm$ 78 pl, the Brunauer-Emmett-Teller (BET;

Brunauer et al., 1938) specific surface area of the fine CFA fraction $A_{\text{BET}}$ (see Sec. S8), and the concentration of CFA in suspension $C$. The $n_{\text{s}}$ error was estimated by propagating the uncertainties of $V_{\text{drop}}$, $A_{\text{BET}}$, and the Poisson distribution of particles in suspension.





$$n_{\mathrm{s}}(T) = -\frac{\ln(1 - f_{\mathrm{ice}}(T))}{V_{\mathrm{drop}} \cdot A_{\mathrm{BET}} \cdot C} \tag{2}$$

## 2.4 Sample characterization

In the following we describe the instrumentation used for the analysis of chemical composition, morphology, and crystallogra-
phy of the CFA samples. The most important findings, which will be referred to in discussion of the immersion freezing results
(see Sec. 3), are summarized. For more details concerning sample characterization, see the Supplementary Information.

### 2.4.1 Size selected CFA

**ALABAMA**

ALABAMA, which was originally developed for aircraft operation (Brands et al., 2011) but is also used in ground-based
campaigns (Roth et al., 2016; Schmidt et al., 2017), is a single particle laser ablation instrument using a Z-shaped time-of-flight
mass spectrometer. After entering the instrument, aerosol particles are focused to a narrow particle beam by an aerodynamic
lens. Subsequently, the particles pass two detection lasers (405 nm) which deliver information about the particle time-of-
flight and hence about the particle vaccum aerodynamic diameter for particles in a size range between ~200 and 1000 nm.
In addition, the time information is necessary to trigger the ablation laser. The latter, a Nd:YAG operating at a wavelength
of 266 nm, evaporates the particles and ionizes the molecule fragments. The resulting ions are analyzed in the bipolar mass
spectrometer such that one anion and one cation spectrum is obtained for each single particle, yielding information about their
chemical composition. Single particle mass spectra were averaged, resulting in a mean chemical composition of each CFA
sample from both dry and wet particle generation.

The overall composition of the CFA samples is comparable to mineral dust, as elements like Al, Ca, K, Fe, Si, S, P, Na,
and Mg frequently occur. Some trace elements seem to be characteristic for the sampled CFA particles. Especially Ti-, Sr-,
and Ba-related mass-to-charge ratios occur in more than 50 % of all dry sampled particles (see Fig. S1 and S2) and could
potentially be used as indicators for CFA particles in the atmosphere, together with the overall fingerprint of their mass spectra.
There are some features, which could explain differences in immersion freezing behavior between the different samples (see
Fig. S4). For example, CFA1 contains most Ca and S in 300 nm particles and CFA3 contains least Ca and S in 300 nm particles,
whereas CFA3 contains most Si in 300 nm particles and CFA1 contains least Si 300 nm particles. Furthermore, CFA1 is the
sample with the highest amount of Pb in 300 nm particles. The comparison of averaged mass spectra of dry- and wet-generated
CFA particles indicates the hydration of oxides, e.g., CaO, SrO, and BaO, in suspension (see Fig. S3). S-containing substances
are present to a lower extent in wet-generated CFA compared to dry-generated particles. A more detailed description of the
ALABAMA results is given in Sec. S1.





**MINI - ESEM/EDX**

Particles were collected on boron substrates with the MINI (Ebert et al., 2016), using one stage with $D_{50} = 1$ $\mu$m. Sampling durations ranged from 30 s to 6 min, depending on average particle number concentrations of the different samples (80 to 300 cm$^{-3}$). Chemical composition, size, and morphology were investigated with a Quanta FEG 400 ESEM. Particles impacted on the boron substrate located in the impaction spot were randomly selected for analysis. Chemical elements with an atomic number larger than 5 were detected with an EDX detector and analyzed with the Oxford software AZtec (version 3.3 SP1). All measurements were carried out with 12.5 keV, 10 mm working distance, and 20 s acquisition time per particle.

CFA1 is the only sample for which a clear difference could be seen between dry and wet particle generation. Whereas dry-generated particles are irregularly shaped agglomerates of small spherules, wet-generated particles occur as both spherules and needle-shaped crystals (see Fig. S5). CFA1 is the only sample where needles were observed in connection with wet particle generation. The major elements detected by EDX agree with the ones identified in the ALABAMA mass spectra (see Table S1). However, trace elements, e.g., Ti, Sr, and Ba, could not be found, presumably for statistical reasons. A more detailed description of the SEM/EDX results is given in Sec. S2.

**2.4.2 Bulk CFA**

**XRD**

For crystallographic characterization of the CFA samples, X-Ray Diffraction (XRD) analyses were performed on both, dry particles and suspension particles. Dry particles were ground using mortar and pestle before being pressed into a sample holder as densely as possible. CFA suspensions were prepared as for the LACIS measurements (see Sec. 2.2.2) and then left in a desiccator (steady flow of particle free, dry air) until all water was evaporated. The remaining dry powder was pressed into a sample holder. Both procedures were applied to all four samples, resulting in eight measurements. A Bragg-Bentano diffractometer with a Cu anode (Philips X'Pert) was used to perform 2Theta-Omega scans from 10° to 70° with a step size of 0.03° and an integration time of 20 s. Quantitative phase identification was done by Rietveld refinement using reference patterns from the Crystallography Open Database (Gražulis et al., 2009).

The XRD patterns indicate quartz (SiO$_2$) as the major crystalline phase in all CFA samples (see Fig. S6 to S9). Furthermore, anhydrite and lime occur in all samples, but to the largest extent in CFA1. CFA1 is also the only sample, where a definite change can be seen between the original dry sample and the sample that was produced by evaporating all water from the suspension. Here, the conversion of anhydrite (CaSO$_4$) to gypsum (CaSO$_4$·2H$_2$O, see R1) and the conversion of lime (CaO) to calcite (CaCO$_3$, see R2 and R3) can be observed. CFA3 is the sample with the highest amorphous, i.e., non-crystalline, fraction in bulk, and likely also in 300 nm particles, as an increase of the amorphous fraction in CFA towards smaller particle sizes has been reported in a previous study (Matsunaga et al., 2002). A more detailed description of the XRD results is given in Sec. S3.

$$CaSO_4 + 2H_2O \rightarrow CaSO_4 \cdot 2H_2O \tag{R1}$$





$$CaO + H_2O \rightarrow Ca(OH)_2 \tag{R2}$$

$$Ca(OH)_2 + CO_2 \rightarrow CaCO_3 + H_2O \tag{R3}$$

**Chemical composition**

The bulk chemical composition analysis was performed using Inductively Coupled Plasma-Sector Field Mass Spectrometry (ICP-SFMS) at ALS Scandinavia AB (Luleå, Sweden). Measured mass fractions of major ions were recalculated into their most common oxide forms (see Fig. S10). Because of its high CaO content of 26 wt%, CFA1 is classified as class C CFA according to the American Society for Testing Materials (ASTM, standard C618, 2015). Class C CFA is cementitious, i.e., self-hardening in contact with water, and the occurrence of needles in wet-generated CFA1 particles could cause this cementitous property. CFA2, CFA3, and CFA4 are class F CFA, meaning that additives are needed to induce hardening of a CFA-water mixture. A more detailed description of the bulk chemical composition results is given in Sec. S4.

In addition to the ICP-SFMS measurements, water activity and pH values of CFA suspensions were determined. The water activity of the CFA samples was ~1, i.e., no difference to pure water could be detected. The CFA2, CFA3, and CFA4 suspensions were neutral to slightly alkaline (pH~7-8). The CFA1 suspension was strongly alkaline (pH~11), likely due to the high CaO content and the formation of portlandite ($Ca(OH)_2$, see R2) which dissociates into $Ca^{2+}$ and $OH^-$ ions.

## 3   Results and discussion

In the following, refer to Fig. 2 for comparing LACIS measurements of individual CFA samples with measurements of different substances contained in CFA. Figure 3 shows SPIN experiments with our CFA samples and U.S. American CFA samples (Garimella, 2016). Figure 4 shows the comparison of CFA results from LACIS, LINA, and WISDOM and the intercomparison between samples.

### 3.1   Dry particle generation

#### 3.1.1   CFA

LACIS measurements with dry-generated CFA particles are reported between -26°C, where the first signal above the limit of detection could be observed, and -37°C, where homogeneous ice nucleation starts to contribute. Data showing measurements with dry-generated particles from CFA1 have previously been published in Grawe et al. (2016). Comparing the $n_s$ spectra of all four CFA samples (see full circles in Fig. 4) shows variation within a factor of 37 (difference between CFA2 and CFA3 at -28°C). CFA1 has the highest $n_s$, followed by CFA2, CFA4, and CFA3. This order is valid throughout the whole examined temperature range, except for $T > -29°C$, where $n_s$ decreases rapidly in case of CFA1. The curve shape for $T < -29°C$ with the relatively shallow increase is comparable for all samples. The broad temperature range, in which the increase in $n_s$ is





observed, hints at inhomogeneous ice nucleation properties, i.e., different types of ice nucleation active sites at the surface of the CFA particles.

To put the efficiency of the CFA particles into perspective, Fig. 2 includes fits to LACIS measurements with a K-feldspar sample (76 % microcline, 24 % albite) and different kinds of mineral dust which featured a similar immersion freezing behavior

after coating with sulfuric acid (clay mineral baseline) by Augustin-Bauditz et al. (2014). Dry-generated CFA particles are not as efficient as the K-feldspar sample, which is also the most efficient mineral dust sample investigated with LACIS so far, but CFA1 is only one order of magnitude below. All of our dry-generated CFA samples are at least one order of magnitude above the clay mineral baseline. In conclusion, the dry-generated CFA particles are comparable to mineral dust in their immersion freezing behavior.

Figure 3 shows a comparison of SPIN measurements with 300 nm CFA particles between this study and Garimella (2016), who performed immersion freezing measurements with four U.S. American CFA samples, two class C and two class F samples. Only towards the warmer end of the examined temperate range, $n_s$ of the U.S. American samples is comparable to what we found for the German ones. At -36.5 °C, the lowest temperature at which both instruments have been operated, $n_s$ of the U.S. American samples is up to two orders of magnitude lower than $n_s$ of the German samples. In general, the $n_s$ spectra of the

U.S. American samples have a much shallower slope than the German CFA $n_s$ spectra. As the same type of instrument was used for both investigations, we conclude that differences between the German and U.S. American CFA samples originate from differences in physico-chemical particle properties, and not from differences in methodology. Both SPIN experiments, ours and that of Garimella (2016), have in common that no large inter-sample variability was observed. This is in contrast to LACIS, where the class C CFA (CFA1) clearly has the highest efficiency. SPIN results have earlier been shown to differ from

results obtained with instruments specially developed to measure immersion freezing (DeMott et al., 2015; Burkert-Kohn et al., 2017). See Sec. 4 for details on the intercomparison between SPIN and LACIS in the framework of the present study.

### 3.1.2  Comparison of CFA with anhydrite, lime, and quartz

From the comparison of $n_s$ to chemical information from ALABAMA measurements, it was concluded that components containing Ca and S could contribute to the observed differences in immersion freezing behavior between the CFA samples

(see Fig. S4 a). The occurrence of the Ca cation cluster series ($(CaO)_n$, $(CaO)_nH$, and partially $Ca(CaO)_n$) together with the S anion cluster series ($SO_n$, see Fig. S1) could be an indication for the presence of anhydrite, as suggested by Gallavardin et al. (2008). Therefore, anhydrite, and also lime, were chosen as test substances for additional LACIS measurements. To our knowledge, these are the first immersion freezing measurements using dry-generated anhydrite and lime particles. Both substances are known to occur in CFA and are enriched in submicron CFA particles (Enders, 1996; Querol et al., 1996).

Anhydrite is of special interest because Havlíček et al. (1993) found that water soluble material on the CFA particle surface is mainly anhydrite and suggested that it is responsible for initiating heterogeneous freezing on the particles.

Both anhydrite and lime are efficient INPs in the immersion mode when being dry-generated (see Fig. 2). Note that multiple charge correction was not possible for LACIS measurements with anhydrite and lime (in contrast to CFA). The correction would shift the $n_s$ spectra of anhydrite and lime towards lower $n_s$ values but the slope would stay the same. Generally, multiple



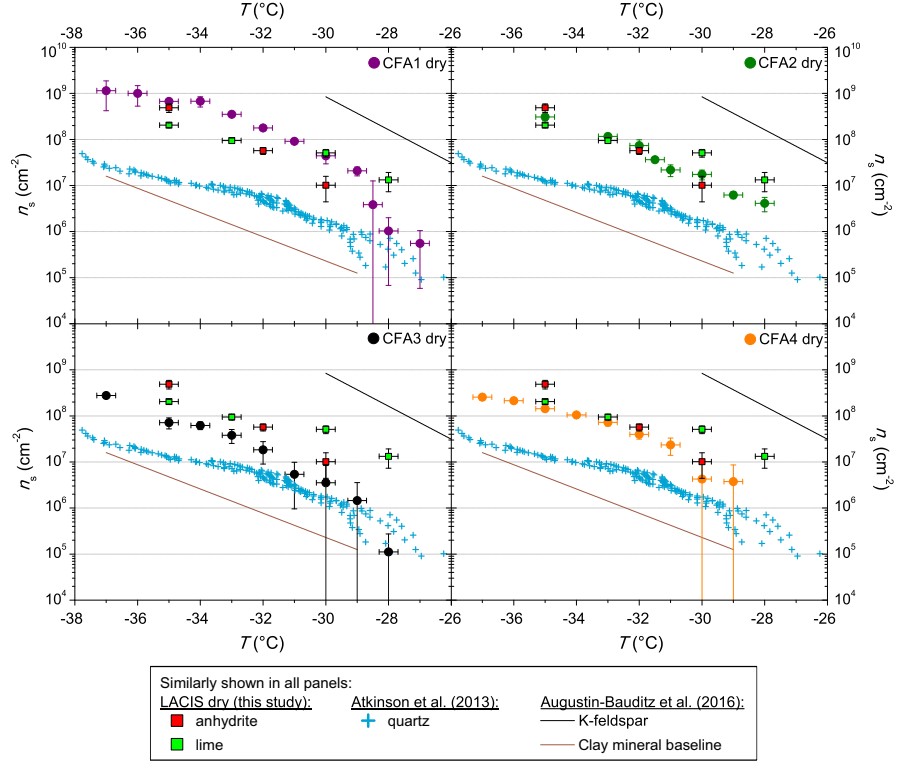

**Figure 2.** $n_s$ from LACIS measurements with dry-generated CFA particles. Measurements with dry-generated anhydrite and lime are included for comparison in all panels but are (in contrast to CFA) not corrected with respect to multiple charges. Measurements with quartz shown in all panels are taken from Atkinson et al. (2013). Fit lines to LACIS measurements with a K-feldspar sample and different kinds of mineral dust coated with sulfuric acid (clay mineral baseline) are taken from Augustin-Bauditz et al. (2014).

charge correction lowers $n_s$ values for the dry-generated CFA particles by less than factor 3.5 and we expect that it would be comparable for the anhydrite and lime particles. Anhydrite is more efficient than lime at $T = -35°C$ (factor 2) and less efficient at $T = -30°C$ (factor 5), i.e., there is a slightly steeper slope of the anhydrite $n_s$ spectrum. Shape and magnitude of the anhydrite $n_s$ spectrum are comparable within measurement uncertainty to what was found for CFA2 and CFA4. CFA3,

5  which contains least Ca and S, and presumably least anhydrite in 300 nm particles, is less efficient than pure anhydrite. CFA1 is more efficient than pure anhydrite, indicating that other compounds might influence the immersion freezing efficiency of this sample. A possible component contributing to $n_s$ of CFA1 might be Pb, which occurs in 20 % of 300 nm particles from CFA1 (in $\leq$10 % of particles from CFA2, CFA3, and CFA4, see Fig. S2) and has been discussed previously as potential INP, or as amplifying the ice nucleation efficiency of other compounds (Cziczo et al., 2009; Kamphus et al., 2010).

10  Quartz, which is the main crystalline phase of all our CFA samples according to XRD measurements and likely also occurs in 300 nm CFA particles (Si was identified by both ALABAMA and ESEM/EDX), is at least one order of magnitude less efficient than CFA1, CFA2, and CFA4. We compare the CFA results to cold stage measurements by Atkinson et al. (2013) here, because



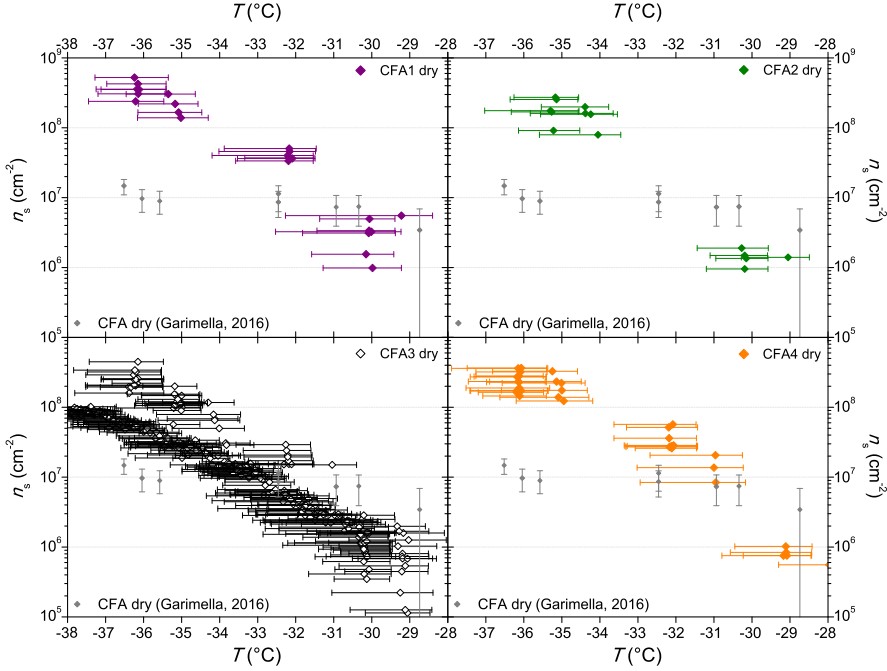

**Figure 3.** $n_s$ from SPIN measurements with dry-generated 300 nm CFA particles. Measurements with U.S. American CFA samples (Garimella, 2016) are included for comparison.

this data set spans the relevant temperature range and $n_s$ values of dry-generated quartz particles from LACIS measurements (not shown here) are comparable to Atkinson et al. (2013). This could be an indication for the immersion freezing behavior of quartz being independent of particle generation technique and measurement method. $n_s$ of CFA3, which contains most Si (presumably quartz) in 300 nm particles is higher by factor 2 to 10 compared to the quartz $n_s$ spectrum. This indicates that

5   quartz might contribute to some of the observed immersion freezing behavior, especially in the case of CFA3, but it is not the most active component in CFA1, CFA2, and CFA4. ALABAMA results show that 300 nm particles in CFA1 contain least Si compounds, followed by CFA2 and CFA4, which supports this hypothesis (see Fig. S4 b).

The hypothesis that the amorphous material in CFA has a promoting effect on its immersion freezing efficiency (Umo et al., 2015) cannot be confirmed for our samples. XRD investigations show that CFA3, which was the least efficient of the four

10  dry-dispersed samples, contains the highest amorphous fraction.

### 3.2   Suspension methods

#### 3.2.1   CFA

Figure 4 summarizes $n_s$ derived from LACIS measurements with dry- and wet-generated CFA particles (full and open circles), and $n_s$ from WISDOM measurements with the fine CFA fraction (squares) and LINA measurements with the bulk CFA (tri-





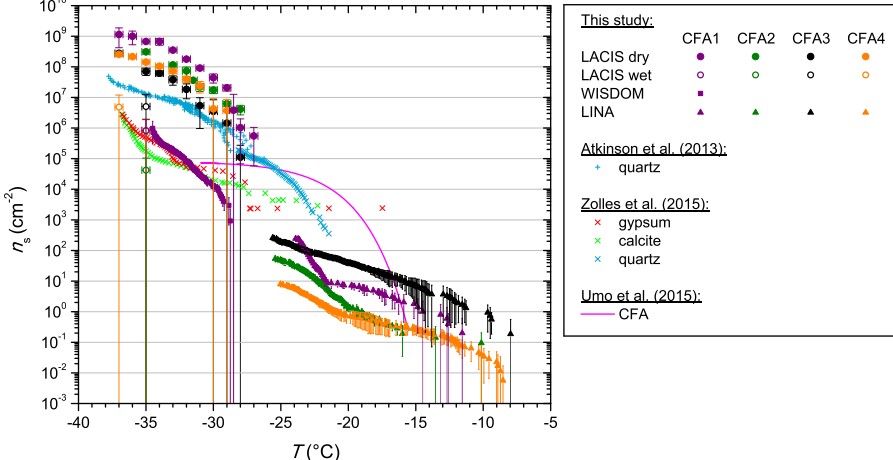

**Figure 4.** $n_s$ from LACIS measurements with dry- and wet-generated 300 nm particles. $n_s$ from WISDOM measurements with the fine CFA fraction and $n_s$ from LINA measurements with bulk material are included for comparison. Measurements with gypsum, calcite, and quartz are taken from Zolles et al. (2015) and Atkinson et al. (2013), measurements with a CFA sample of different origin from Umo et al. (2015).

angles). Firstly, LACIS results will be described and then compared to those of the other two instruments. Secondly, we will compare to measurements with the hydration products of anhydrite and lime by Zolles et al. (2015), and finally to measurements with a CFA sample of different origin by Umo et al. (2015). A comparison to measurements by Havlíček et al. (1993) is unfeasible because no specific surface area values of the samples are given in this publication.

## 5  LACIS

When comparing $n_s$ from LACIS measurements with dry-generated (full circles in Fig. 4) to measurements with wet-generated particles (open circles), a significant decrease can be seen. $n_s$ was lowered by between one (CFA3) and four (CFA2) orders of magnitude at -35°C. $n_s$ values of wet-generated particles vary by up to two orders of magnitude between the four CFA samples. This can possibly be attributed to low values of $f_{ice}$ which are only slightly above values usually measured for homogeneous nucleation (see Fig. B1), i.e., close to the limit of detection. As a result, the error in $f_{ice}$ and $n_s$ is larger than for the dry-generated particles at the same temperature.

Note that data for wet-generated CFA1 particles differ from those published in Grawe et al. (2016), which needed to be corrected due to the identification of a sample-specific artifact (see Appendix B). ESEM images from both, Grawe et al. (2016) and the present study, show two different particle types sampled after wet particle generation and size selection of CFA1, i.e., spheroidal particles and needle shaped crystals (see Fig. S5).

The occurrence of needles suggests that compounds are dissolved from the particles in suspension. During LACIS measurements, purely water soluble particles would activate to droplets which then would only freeze homogeneously, causing an underestimation of $f_{ice}$. To make sure that no purely water soluble particles with a size of 300 nm were produced when




spraying the suspensions, size distributions of particles from all CFA suspensions were measured before LACIS experiments took place. From the size distribution measurements (see Sec. S5), we conclude that a negligibly small number of purely water soluble particles with a size of 300 nm was produced from CFA2, CFA3, and CFA4, i.e., the decrease we observe in transition from dry to wet particle generation is not caused by a measurement artifact. The evaluation of the CFA1 size distribution is not
unambiguous because of the superimposition of size distributions of spheroidal and needle shaped particles.

A decrease in immersion freezing efficiency from dry to wet particle generation was already reported for CFA and coal bottom ash in Grawe et al. (2016). There, it was hypothesized that the increased time that the particles spend in contact with water leads to a change in physico-chemical particle properties which then causes the observed decrease. At this point, it was not possible to identify relevant processes because information on the chemical composition of 300 nm particles was missing.
In the framework of the present study, differences in chemical composition of dry- and wet-generated CFA particles were identified (see Sec. 2.4 and S1), and will be discussed in relation with the immersion freezing results in Sec. 3.2.2.

**Comparison of LACIS, LINA, and WISDOM results**

LINA measurements (triangles in Fig. 4) were performed between 0 and -26°C. In the temperature range from -8 to -23°C, CFA3 has the highest, and CFA4 the lowest $n_\mathrm{s}$ values of all samples, with those being two orders of magnitude apart. CFA1
shows a steep increase in $n_\mathrm{s}$ between -21 and -24°C, below which all droplets were frozen. In contrast, the last droplets of CFA2, CFA3, and CFA4 suspensions froze below -25°C.

WISDOM measurements (squares in Fig. 4) were performed as an attempt to close the temperature gap between LACIS measurements with wet-generated particles ($T \leq -35°C$) and LINA measurements ($T \geq -26°C$). This could not be realized for two reasons. Firstly, WISDOM measurements with 0.1 wt% suspensions were only possible with CFA1, because the
other samples showed no immersion freezing activity. Increasing the concentration to a level, for which signals above the homogeneous freezing limit could be expected, led to strong settling of particles in the CFA2, CFA3, and CFA4 suspensions. Secondly, freezing was only observed for $T \leq -29°C$ for CFA1, i.e., there is no temperature overlap between LINA and WISDOM. However, extrapolation suggests that both instruments could yield similar results.

Good agreement can be observed for WISDOM and LACIS at $T \approx -35°C$ with CFA1. Firstly, this implies that there is no
pronounced effect of size-dependent composition on the immersion freezing behavior of CFA1. This finding could be specific to CFA1, as it is in contrast to Garimella (2016), who found that $n_\mathrm{s}$ increases with decreasing particle size. Secondly, the good agreement between LACIS and WISDOM indicates that drying of the CFA1 suspension droplets after atomization (which does not take place in WISDOM experiments) does not have a strong effect on the immersion freezing efficiency of the CFA1 particles.

**Comparison to Umo et al. (2015)**

Cold stage measurements with a CFA sample of different origin by Umo et al. (2015; see Fig. 4) yielded results that differ substantially from what we measured in the framework of the present study. The efficiency of the sample investigated by Umo et al. (2015) increases strongly for $-15°C < T < -20°C$ and levels off for $T < -25°C$. This is in contrast to the gradual




increase over a broad temperature range that we observed for our samples. Our suspensions were prepared in the same way as described by Umo et al. (2015), and LINA and the micro-litre Nucleation by Immersed Particles Instrument ($\mu$L-NIPI; Whale et al., 2015) used by Umo et al. (2015) have successfully been intercompared with a different ash sample (not shown). Therefore, we infer that the CFA samples are really different in their immersion freezing behavior and we do not observe

artifacts related to methodology. The comparison to Umo et al. (2015), and the results by Garimella (2016) shown in Fig. 2 and 3, suggest that CFA samples from different geographical origin show a highly variable immersion freezing behavior.

### 3.2.2 Comparison of CFA with gypsum and calcite

A comparison of ALABAMA measurements of dry- and wet-generated CFA particles hints at hydration of several oxides (see Fig. S3). It is difficult to say which hydration reactions in the complex mixture cause the decrease in immersion freezing

behavior in measurements with the suspension methods. However, for bulk CFA1 there is clear evidence from XRD measurements that anhydrite and lime, which were already identified as species potentially influencing immersion freezing of the dry-generated particles, are hydrated in suspension, resulting in the formation of gypsum and calcite (see Fig. S6). In the following, we hence discuss the comparison between CFA suspension particles to measurements presented in Zolles et al. (2015) of gypsum and calcite (see Fig. 4).

Both hydration products, i.e., gypsum and calcite, are lower in their immersion freezing efficiency by three orders of magnitude compared to anhydrite and lime, i.e., as for CFA, there is a significant decrease in efficiency of the hydration products compared to their anhydrous precursors. In general, gypsum and calcite are similar in their immersion freezing efficiency, and LACIS measurements with wet-generated CFA and WISDOM measurements agree within one order of magnitude. The only exception to this is CFA3 which will be discussed below in relation with quartz.

The hydration of anhydrite inevitably takes place once CFA comes into contact with water, because anhydrite is present at the particle surface (Enders, 1996). Sievert et al. (2005) describe the hydration of pure anhydrite particles in the following way: Firstly, anhydrite is dissolved from the particles and $Ca^{2+}$ and $SO_4^{2-}$ ions are hydrated in the solution. The hydrated ions are then adsorbed to the surface of the anhydrite particles due to electrostatic attraction. From this point on, further dissolution and interaction of water molecules with the anhydrite surface is reduced because of the adsorbed layer of hydrated ions. Secondly,

as the thickness of the adsorbed layer increases, cracks are formed through which water molecules migrate to the anhydrite surface. Only then, nuclei of gypsum are formed and crystallization takes place. The first process (formation of adsorbed layer of hydrated ions) is thought to happen rather quickly, the second process (formation of gypsum) can take several hours up to days. See Sec. 3.3 and 4 for details on the duration of hydration.

The formation of calcite occurs via the hydration of lime to portlandite ($Ca(OH)_2$) which is then carbonated (see R2 and

R3). It is possible that this process causes the precipitation of needles in suspension, but only if the lime content is sufficiently high, as for CFA1. It cannot be ruled out that calcite is also formed in the other CFA suspensions, however, in contrast to CFA1, calcite could not be clearly identified in the other samples by XRD.




Possibly, both above described mechanisms, and potentially even more hydration reactions, cause the observed decrease in immersion freezing efficiency in transition from dry to wet particle generation. Additional LACIS measurements with different sample treatments were performed to verify this hypothesis (see Sec. 3.3).

### 3.2.3 Comparison of CFA with quartz

In addition to quartz measurements by Atkinson et al. (2013), we now include quartz measurements by Zolles et al. (2015) in our discussion because they cover $T > -28°C$, which is the more relevant temperature range for our cold stage measurements with CFA. We compare to the most efficient of the quartz samples investigated by Zolles et al. (2015). The $n_s$ spectra of the quartz samples used by Zolles et al. (2015) and Atkinson et al. (2013) agree in the narrow temperature overlap ($-26°C < T < -28°C$). It is obvious that quartz is significantly more efficient in the immersion mode than suspended particles of CFA1, CFA2, and CFA4, with $n_s$ being at least one order of magnitude higher over the complete investigated temperature range. The deviation is smallest for CFA3 which contains most Si species (quartz) and least Ca and S species in 300 nm particles. For this sample, we assume the smallest effect of the hydration reactions and a larger influence of quartz on the immersion freezing behavior compared to the other samples. The fact that the other samples also contain significant amounts of quartz, both in 300 nm particles and in bulk, and, nevertheless, feature a much lower efficiency, supports the hypothesis of the particles being covered by a layer. In case of dry particle generation, the layer is more efficient at initiating immersion freezing than quartz. In case of the suspension methods, the layer is less efficient than quartz, with this change brought on by the above described hydration reactions.

### 3.3 Effect of sample treatment on the immersion freezing efficiency of CFA

Additional LACIS measurements with differently treated CFA and anhydrite samples, as well as pure gypsum, were performed in order to test the hypothesis that the hydration of anhydrite leads to a decrease of immersion freezing efficiency in suspension (see Fig. 5). All measurements were performed at -35°C with 300 nm particles. Here, we forewent the multiple charge correction for better comparability to measurements that took place after the campaign where no correction was possible. The corrected values for CFA (used for the $n_s$ calculations in Fig. 4) are shown as circles.

When comparing dry-generated CFA particles with wet-generated particles, either from a fresh suspension (i.e., measured within 5 min after preparation), or from the standard suspension (10 min of ultrasonification and 24 h stirring), a decrease in $f_{ice}(-35°C)$ can be observed. However, the particles from the freshly prepared suspension seem to be slightly more efficient than the ones from the standard suspension. The only exception is CFA3, where it was extremely difficult to generate a sufficiently high particle number concentration from the fresh suspension, resulting in a large error due to the small amount of classified hydrometeors (~500). Dry- and wet-generated anhydrite particles show the same trend as observed for CFA, i.e., the wet-generated particles are significantly less efficient than the dry-generated particles, and the longer the particles stay in suspension, the stronger the decrease in $f_{ice}$.

Sullivan et al. (2010) describe an increase in hygroscopicity of wet-generated anhydrite particles in comparison to dry particle generation. Also, the hygroscopicity of the wet-generated particles increased with the time that the particles spent in



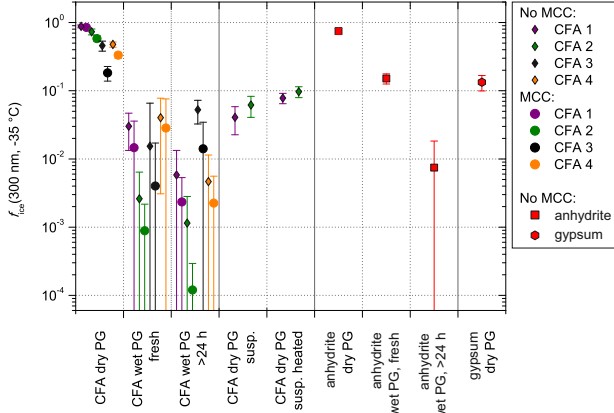

**Figure 5.** $f_{ice}(-35°C)$ from LACIS measurements with 300 nm particles. Multiple Charge Correction (MCC) was not performed, except for the measurements indicated by the circles in the first three columns. "Dry PG susp." means dry Particle Generation (PG) with a sample resulting from the evaporation of a suspension and "dry PG susp. heated" means additional heating of this sample to 250°C prior to particle generation.

the suspension. Sullivan et al. (2010) attribute this behavior to the formation of hydrates and hypothesize that this process could have an effect not only on hygroscopicity but also on the ice nucleation efficiency of the particles. In our case, the dependency of immersion freezing efficiency on suspension time could result from the two stages of anhydrite hydration described in Sec. 3.2.2. Firstly, on the time scale of minutes, anhydrite is dissolved and hydrated ions form a layer on top of the CFA particles

causing a sudden decrease in immersion freezing efficiency. It seems that the limited suspension time of 1.6 s in case of dry particle generation is not sufficient to cause hydration. Secondly, on the time scale of several hours up to days, anhydrite is converted into gypsum which decreases $f_{ice}$ further. Gypsum, like anhydrite, consists of molecules which are strong electrical dipoles (Klimchouk, 1996) and hence will also be surrounded by hydrated ions in suspension.

    The ultrasonicated and stirred suspensions of CFA1 and CFA2 were left in a desiccator (steady flow of particle free, dry air)

until all water was evaporated. XRD measurements of the resulting powder show that the anhydrite-gypsum conversion had taken place and we assume that gypsum was already present in the stirred suspensions. The powder was then dry-dispersed and $f_{ice}(-35°C)$ of 300 nm particles was measured. An increase of almost one order of magnitude for CFA1 and almost two orders of magnitude for CFA2 in comparison to the wet-generated particles was registered. We attribute this increase to the difference in particle generation. For wet particle generation, possibly only the bulk water is removed in the diffusion dryer

downstream of the atomizer, whereas the water molecules in the layer of hydrated ions remain. Drying in a desiccator, which takes several days, could lead to partial dehydration, i.e., removal of the hydrated layer surrounding the CFA particles. For dry particle generation, limited suspension time of 1.6 s in LACIS is apparently not long enough for rehydration.

    Additionally, the powder from the evaporated suspensions of CFA1 and CFA2 was heated to 250°C for 15 min. According to Deer et al. (1992), this temperature is sufficient to dehydrate gypsum and form anhydrite. $f_{ice}(-35°C)$ of 300 nm particles





slightly increased by a factor of 2 after the heat treatment, but it did not restore the immersion freezing efficiency of the original dry-dispersed particles. It is known that other hydrated species are present in the suspension particles that are only dehydrating at much higher temperatures (e.g., dehydration temperature of portlandite: 510°C; Bai et al., 1994) and hence it is not surprising that only a small increase in $f_{ice}$ could be achieved. In general, there is good agreement between measurements

with dry-generated gypsum particles (see Fig. 5) and particles from the dried and heated CFA suspensions, indicating that gypsum is present in the investigated 300 nm CFA particles after they have spent a sufficiently long time in water.

It is beyond the scope of this paper to examine why hydration leads to a lower immersion freezing efficiency. Hence, we only offer some possible explanations here without further discussing their likelihood. A simple explanation would be that the adsorbed layer of hydrated ions on the particle surface blocks the interaction with the surrounding water molecules.

Consequently, freezing would not be triggered as efficiently as for the dry-generated particles, where the contact with water is too short to dissolve a sufficient amount of anhydrite. Another hypothesis (Sihvonen et al., 2014) describes a change in lattice parameters upon forced hydration of mineral dust particles towards a greater mismatch with ice.

## 4   Atmospheric implications

In view of the atmospheric relevance of the above described findings, it is important to discuss, whether the observed decrease

in immersion freezing efficiency of CFA associated with switching from dry to wet particle generation would also occur in the atmosphere. From LACIS measurements with the freshly prepared CFA suspensions, we know that particles in the bulk suspension are deactivated within ~5 min but it is not clear if this would also be observed when a single particle is immersed in a cloud droplet. As already mentioned, it seems that 1.6 s, which is the residence time of CFA particles in water for LACIS measurements with dry-generated CFA, are not enough to cause hydration of anhydrite and lime.

Unfortunately, an increase in nucleation time by more than a factor of 2 is not possible with LACIS. However, during the campaign, SPIN measurements with dry-generated CFA particles were performed above water saturation (see Appendix C). From the results, which agree with LACIS for CFA3 and CFA4 but are below LACIS for CFA1 and CFA2, it can be speculated that the longer residence time in SPIN (~10 s) already leads to some deactivation by the formation of a hydrated layer on top of those particles which contain most water soluble anhydrite. However, the effect is much more pronounced for longer hydration

times, as can be seen in the results of SPIN measurements with wet-generated particles. It would be necessary to increase nucleation time further to evaluate the time needed to decrease the immersion freezing efficiency of single particles immersed in droplets to the efficiency of particles hydrated in the bulk suspension. Within the framework of the present study, it was not possible to keep cloud droplets with a single immersed CFA particle stable for longer than a few seconds before investigating immersion freezing. Hence, we can only give a range of how efficiently CFA induces immersion freezing in the atmosphere,

because, for CFA containing a certain amount of anhydrite, this will depend on the time between activation to cloud droplets and triggering of freezing.

Figure 6 shows INP concentrations estimated from our CFA measurements in combination with size distributions measured ~80 km downstream of a coal-fired power plant (Parungo et al., 1978). For this, the the ambient size distribution was subtracted





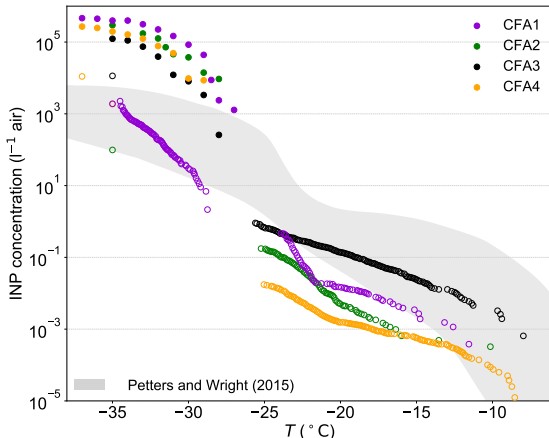

**Figure 6.** Estimated INP concentrations ~80 km downstream of a coal-fired power plant based on size distributions measured in a plume by Parungo et al. (1978). Full circles represent dry-particle generation, open circles wet-particle generation and cold stage measurements. The shaded area indicates typical atmospheric INP concentrations derived from precipitation samples (Petters and Wright, 2015).

from the size distribution in the plume to only consider particles emitted from the power plant. The procedure is explained in detail in Grawe et al. (2016), where we already estimated the INP concentration due to the emission of CFA1 to be higher than typical atmospheric INP concentrations, assuming $n_s$ of dry-generated particles. This is equally true for the other dry-generated CFA samples, with estimated INP concentrations being roughly two orders of magnitude above the upper boundary given by

Petters and Wright (2015) at -37°C. In light of our new findings, and assuming that atmospheric processing will lead to a decrease in immersion freezing efficiency, we also estimated the INP concentrations using $n_s$ from LACIS measurements with wet-generated CFA and cold stage measurements. Above -30°C, INP concentrations derived from measurements with CFA from suspension are close to or below the lower boundary given by Petters and Wright (2015), except for CFA3, which is within the boundaries given by Petters and Wright (2015) for $T > -23$°C. This indicates that the majority of our CFA samples

only contributes very little to atmospheric INP concentrations above -30°C, when we assume that the suspension results are representative for processes occurring in the atmosphere. Our estimate suggests that particles from CFA1, CFA2, and CFA4 only become relevant for atmospheric immersion freezing at temperatures below -30°C. Note that in close proximity to the source, i.e., in an undiluted plume directly after emission, INP concentrations will be much higher than estimated above. At greater distances from the power plant, INP concentrations will be significantly lower due to dilution. Garimella (2016)

estimates that CFA particles are present at cirrus level in concentrations of ~0.1 to 1 l$^{-1}$.





## 5 Summary and conclusions

In the framework of this study, four CFA samples from German power plants were investigated concerning their immersion freezing behavior, chemical composition, morphology, and crystallography. We can now give answers to the following questions from the introduction:

– Do CFA samples from different power plants feature a similar immersion freezing behavior? and

 – Is the deactivation in transition from dry to wet particle generation observable for different CFA samples?

All four samples were found to be efficient INPs in the immersion mode below -28°C. The $n_s$ spectra of dry-generated particles differed by approximately one order of magnitude, with the curve shapes being very similar. A decrease in immersion freezing efficiency was observed for all of our samples when particles were generated from a suspension. However, the data
set is still too small to make a conclusive statement about the variability in immersion freezing results caused by different samples and differences in methodology. Comparisons to samples of different geographical origin (Umo et al., 2015; Garimella, 2016) suggest that the spread is indeed larger than what we found for the German CFA samples. Further immersion freezing measurements with more CFA samples from different sources, which should also focus on the effect of hydration, would be needed to provide a suitable parameterization.

– Is it possible to find a connection between physico-chemical sample properties and the observed immersion freezing behavior?

From ALABAMA measurements it was derived that the amount of molecular species containing Ca and S correlates with the immersion freezing efficiency of the dry-generated samples. Additional LACIS measurements with anhydrite and lime yielded similar results as for CFA, suggesting that both substances contribute to the observed freezing behavior. Both anhydrite and
lime are hydrated (lime also carbonated) in contact with water which might cause a decrease in immersion freezing efficiency. Cold stage measurements with the hydration products gypsum and calcite (Zolles et al., 2015) are comparable to LACIS measurements with wet-generated CFA particles and to WISDOM measurements. An exception is CFA3, which contains least Ca and most Si in both 300 nm particles and bulk. Here, the decrease in immersion freezing efficiency in transition from dry to wet particle generation is smallest, and LACIS measurements are relatively close to cold stage measurements with quartz
(Atkinson et al., 2013). Quartz was detected as the major crystalline phase in all of the bulk samples. From this, we conclude that an influence of quartz on the immersion freezing behavior of CFA can only be seen in case the amount of anhydrite and lime is below a certain, not clearly definable, threshold.

 – Which particle generation technique (dry or wet particle generation) or measurement method (single particle vs. cold stage) is appropriate for representing atmospheric processes after CFA emission?

It is important to know that for CFA, it is necessary to consider dissolution effects in suspension and changes in immersion freezing behavior on short time scales. We observed that dry-generated particles, which were immersed in droplets in LACIS





for 1.6 s before freezing, are efficient INPs and can potentially contribute to the atmospheric INP spectrum if concentrations are high. However, for two of the samples, a decrease in freezing efficiency could already be seen when particles were immersed for ~10 s in SPIN, suggesting that the ability of CFA to act as INP can decrease quickly in contact with water. Estimating atmospheric INP concentrations due to CFA emission, and assuming atmospheric processing of the particles, indicates that

CFA is relevant at $T < -30°C$, i.e., an effect on cirrus formation could be possible. Concerning this, it could also be worthwhile to further investigate deposition nucleation on CFA particles.

An approach to improve process understanding of CFA ageing in the atmosphere is to either sample particles in a coal-fired power plant plume on filters, or perform *in situ* INP measurements, preferably at several distances downstream of the stack. In case it should turn out that the lower limit given by LACIS measurements with wet-generated particles and cold stage

measurements is reproducible, one might be able to provide sample-specific parameterizations and, once more samples have been investigated, boundaries for the immersion freezing efficiency of CFA.

Future research should also focus on quantifying CFA emissions and temporal and spacial variability of CFA particle concentrations. Mass spectrometry measurements of CFA, as performed in the framework of this study, can help to identify CFA in the atmosphere. However, the classification of single particles still remains difficult because CFA particles are heterogeneous in

their composition and not all of them contain a characteristic marker. More composition measurements of atmospheric aerosol and ice crystal residues are needed to better assess the effect of CFA emission on weather and climate.

*Data availability.* The data are available in a publicly accessible MySQL portal at http://imk-aaf-s1.imk-aaf.kit.edu/inuit/ and upon request to the contact author.

*Sample availability.* In consultation with the power plant operators, sample origin shall not be disclosed and distribution is not possible.

**Appendix A: LINA water background correction, $n_s$ calculation, and error estimation**

For each CFA sample, $f_{ice, H_2O}(T)$ was determined from LINA measurements with the distilled water that was used to prepare the CFA suspension. Freezing caused by impurities in the distilled water and on the glass slide was accounted for in the following way: Firstly, the number of sites active at a given temperature $T$ per droplet volume $V_{drop}$, $K_{H_2O}(T)$, was calculated for distilled water (Eq. A1; Vali, 1971). Secondly, this value was subtracted from $K_{CFA}(T)$ (Eq. A2; Umo et al., 2015). Finally,

the difference was used to calculate a corrected $n_s$ value (Eq. A3), with $C$ the mass concentration of CFA in suspension and $A_{BET}$ the BET specific surface area (see Sec. S8).





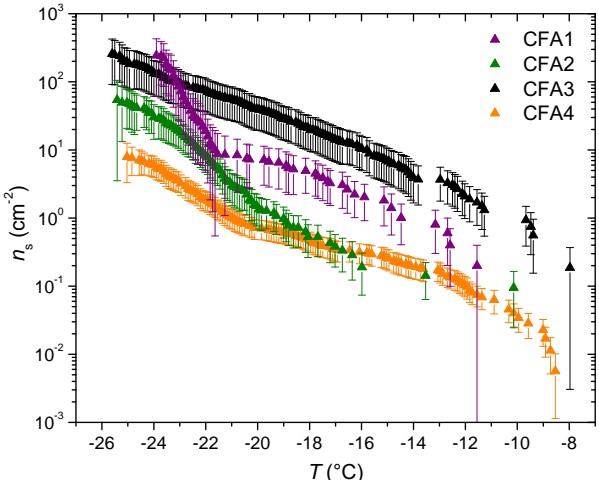

**Figure A1.** $n_s\,(T)$ from LINA measurements. Vertical error bars are the result of propagating uncertainties in weighing, BET surface area, pipette volumes, and distribution of particles in the suspension.

$$K_{\mathrm{H_2O}}(T) = -\frac{\ln\left(1 - f_{\mathrm{ice,\,H_2O}}(T)\right)}{V_{\mathrm{drop}}} \tag{A1}$$

$$K_{\mathrm{CFA}}(T) = -\frac{\ln\left(1 - f_{\mathrm{ice,\,CFA}}(T)\right)}{V_{\mathrm{drop}}} \tag{A2}$$

$$n_{\mathrm{s,\,corr}}(T) = \frac{K_{\mathrm{CFA}}(T) - K_{\mathrm{H_2O}}(T)}{C \cdot A_{\mathrm{BET}}} \tag{A3}$$

$n_{\mathrm{s,\,corr}}$ values from four measurements were averaged for a mean $n_s$ value, i.e., a total number of 360 droplets was investigated for each sample. The uncertainty of the mean $n_s$, given as vertical error bars in Fig. 4, is equal to the standard deviation of the four $n_{\mathrm{s,\,corr}}$ values. The largest possible $n_s$ error of the LINA measurements is illustrated in Fig. A1. For this, the uncertainties in concentration from weighing of the CFA sample and pipetting distilled water, as well as BET specific surface area, and volume of the droplets were propagated. Here, the error in $f_{\mathrm{ice}}$ was assumed to be related to the standard deviation of the Poisson distribution of particles in the suspension.

## Appendix B:  Comment on misinterpreted LACIS measurements with wet-generated CFA1 particles published in Grawe et al. (2016) and correction of those

Data shown in Fig. B1 is taken from Grawe et al. (2016; similar to Fig. 4 d) and shows $f_{\mathrm{ice}}(T)$ for dry- and wet-generated CFA1 particles. Measurements with dry-generated particles are identical to those shown in Fig. 2. Measurements with wet-generated particles from a suspension, prepared as described by Umo et al. (2015), i.e., 10 min of ultrasonification (US) and 24 h of





stirring, suggest that CFA1 retains some activity even when being wet-generated. $f_{ice}$ was found to be around 5 % between -24 and -35°C, indicating no strong temperature dependence.

At this point it was already known that needles are present among wet-generated CFA1 particles. However, it was assumed that the needles are composed of water soluble material which will dissolve once a needle is immersed in a droplet. $f_{ice}$ would
be underestimated due to the occurrence of purely water soluble particles, and according to this hypothesis, $f_{ice}$ was multiplied by a scaling factor of 4.54 (=1/0.22, assuming that only 22 % of the droplets contained an insoluble particle).

Additional measurements were performed with modified suspensions: When the suspension was prepared without US, just stirring, lower $f_{ice}$ values around 1 % at -35°C were observed. When filtering the suspension through a 200 nm syringe filter, $f_{ice}$ was only slightly above values measured for highly diluted ammonium sulphate droplets, i.e., homogeneous nucleation.
ESEM images (see Fig. S5) show, that CFA1 is indeed the only one of the four CFA samples for which needles form during wet particle generation. Optical microscope images of liquid suspension droplets (see Fig. S12) show that the needle shaped particles are even present in the aqueous environment, disproving the earlier hypothesis of water soluble needles. Even though the substrates were loaded after size selection, needles which are much longer than the selected 300 nm can be seen on the ESEM images and could be introduced into LACIS. This is due to the fact that the dynamic shape factor of the needles differs
significantly from unity. The ESEM images suggest that some of the needles are even longer than the usual droplet diameter at the LACIS outlet, which is 5 $\mu$m. This represents a challenge for the optical detection with TOPS-Ice, because the determination of $f_{ice}$ is based on depolarization, and hence largely on the shape of the hydrometeors. In usual LACIS immersion freezing experiments with 300 nm particles, the supercooled liquid droplets are spherical because a sufficient amount of water vapor is provided to form a thick (with respect to the particle diameter) layer of water on top of the particle upon activation. However, if
we imagine an experiment with particles from the CFA1 suspension, the long needles have a much larger surface area that will be covered by water molecules when exposed to same supersaturation with respect to liquid water. As a result, a much thinner water layer is formed which will not be able to "hide" the irregular particle shape. Due to this, there is a fraction of needles longer than 5 $\mu$m causing droplets being non-spherical, yet unfrozen. Consequently, depolarization signals are produced, which are associated with ice particles. This artifact can also be observed at $T > 0$°C and thus we falsely interpreted signals caused
by long needles as frozen droplets in Grawe et al. (2016), overestimating the immersion freezing efficiency of wet-generated CFA1 particles.

To determine the realistic freezing potential of wet-generated CFA1 particles, the suspension was put through a filter (Whatman®, grade 595, 4-7 $\mu$m particle retention) prior to wet particle generation to remove large needles. As a result, experiments could be conducted with 5 $\mu$m sized droplets, which were then spherical when unfrozen. $f_{ice}$ was found to be below 0.1 % at
-35°C, i.e., the wet-generated CFA1 particles are roughly three orders of magnitude less efficient than the dry-generated ones.

Concerning the lower $f_{ice}$ values for particles from the CFA1 suspension without ultrasonification from Grawe et al. (2016), it can be hypothesized that, due to the lack of agitation, less of the material responsible for the needle formation was dissolved from the CFA particles. Consequently, less and/or shorter needles might have formed which would not disturb the spherical shape of the droplets.



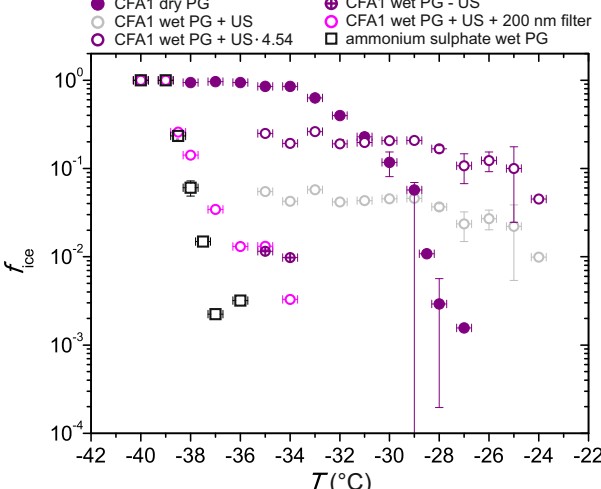

**Figure B1.** Data taken from Fig. 4 d) of Grawe et al. (2016) showing $f_{ice}$ measured with 300 nm dry- and wet-generated CFA1 particles. US: ultrasonification, PG: particle generation.

## Appendix C: Potential influence of residence time on immersion freezing efficiency of dry-generated CFA

SPIN measurements above water saturation ($1.03 \leq S_w <$ droplet breakthrough) were performed with dry-generated 300 nm particles from all four CFA samples and wet-generated 300 nm particles from CFA1 (see Fig. C1). Measurements with wet-generated particles could only be done for CFA1 due to instrument availability. For comparison to LACIS results, SPIN $AF$ data are shown as measured and additionally multiplied by factor 3, based on results from a previous intercomparison campaign including LACIS and SPIN (Burkert-Kohn et al., 2017) and a comparison between a different CFDC and a cloud chamber (DeMott et al., 2015). Corrected data were interpolated for better clarity and are represented by the dashed lines.

The decrease in transition from dry to wet particle generation, which was observed in LACIS, was also measured with SPIN. Concerning dry-generated CFA particles, there is nearly perfect agreement between LACIS and SPIN for CFA3 and CFA4 after correction. In case of CFA2, SPIN results are lower than LACIS results, especially for $T \geq -30°$C. The biggest difference is observed for dry-generated particles of CFA1, where SPIN data is significantly below LACIS for $T \geq -35°$C. A possible explanation could be that CFA1 is the sample with the highest amount of Ca, and presumably anhydrite, that will be dissolved once the particles are activated. The LACIS measurements indicate that an activation time of 1.6 s is too short to cause the formation of an adsorbed layer of hydrated ions (as described in Sec. 3.2.2). The residence time of the particles in SPIN is factor 6 higher and this could be enough time to dissolve a sufficient amount of anhydrite. The dependency of residence time in SPIN on the thermodynamic conditions in the chamber (8.2 s at $T = -30°$C and 9.2 s at $T = -40°$C) could explain why the discrepancy between SPIN and LACIS is higher at higher temperatures. An increase in residence time allows more ions to dissolve from the CFA particle surface at higher temperatures, which consequently leads to a stronger decrease of





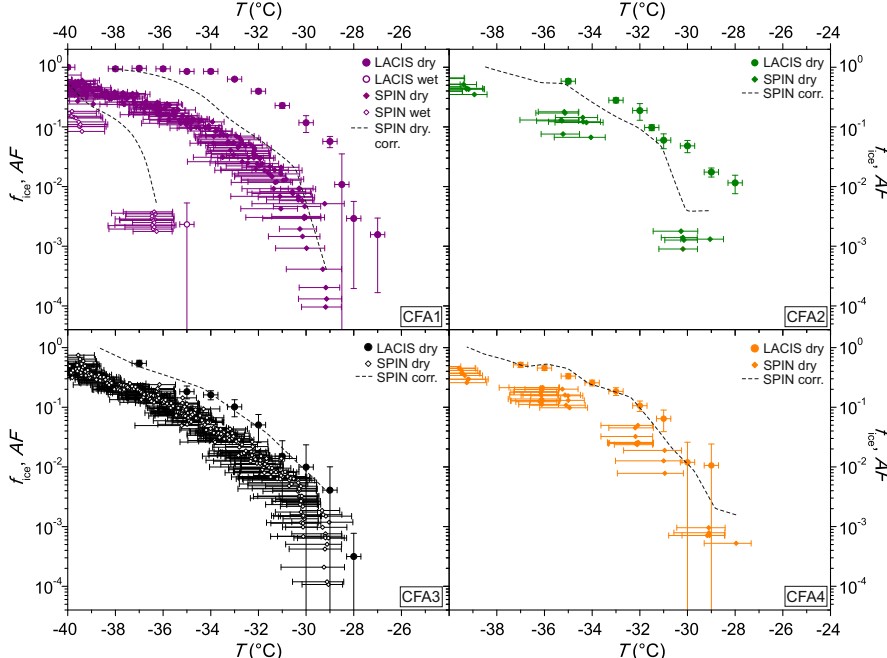

**Figure C1.** Comparison of CFA immersion freezing measurements with SPIN and LACIS (300 nm particles). Dashed lines indicate interpolated SPIN data after a correction factor 3 is applied.

$AF$ at higher temperatures. This effect is also visible for CFA2. Following this hypothesis, CFA3 and CFA4 do not contain a sufficient amount of anhydrite to form a hydrated shell around the particles within ~10 s.

*Author contributions.* S. Grawe wrote the manuscript with contributions from H.-C. Clemen, S. Eriksen-Hammer, N. Reicher, and H. Wex. LACIS measurements and data evaluation were performed by S. Grawe, S. Augustin-Bauditz, and J. Lubitz. LINA measurements and data evaluation were performed by J. Lubitz and S. Grawe. H.-C. Clemen performed ALABAMA measurements and data analysis with the support of J. Schneider. S. Eriksen-Hammer sampled particles with the impactor and performed the ESEM/EDX particle analysis together with M. Ebert. N. Reicher measured with WISDOM and provided BET results. A. Welti performed SPIN measurements and data evaluation. R. Staacke performed XRD measurements. S. Grawe, S. Augustin-Bauditz, F. Stratmann, and H. Wex discussed the immersion freezing results and further experiments after the campaign. H. Wex procured the CFA samples and coordinated the campaign. All co-authors proofread and commented the manuscript.

*Competing interests.* The authors declare that they have no conflict of interests.



*Acknowledgements.* This research was conducted in the framework of the DFG funded Ice Nuclei research UnIT (INUIT, FOR1525), WE 4722/1-2, SCHN1138/2-2. We thank the anonymous suppliers of the CFA samples, M. Sidelmann and M. Bilde (Department of Chemistry, Aarhus University, Denmark) for water-activity measurements, M. Lorenz (Semiconductor Physics Group, University of Leipzig, Germany) for providing access to the XRD instrument, A. Roedger, K. W. Fomba, A. Dietze, S. Fuchs, and D. van Pinxteren (TROPOS, Leipzig, Germany) for bulk chemical composition analysis, X. Gong (TROPOS, Leipzig, Germany) for helpful discussions, R. Heller (Leibniz Institute of Surface Modification, Leipzig, Germany) and J. Voigtländer (TROPOS, Leipzig, Germany) for introduction to the optical microscopes, and T. Conrath (TROPOS, Leipzig, Germany) for technical support.




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
