# Peer review of "Coal fly ash: Linking immersion freezing behavior and physico-chemical particle properties"

_Atmospheric Chemistry and Physics, 2018_

## Referee Comment (RC1) · Anonymous Referee #1 · 25 Jul 2018

The study reported by Sarah Grawe and her co-authors on linking the immersion freezing behaviour of CFA particles to the physicochemical properties of coal fly ash (CFA) is an interesting and timely investigation. For a long time, the study of CFA as ice-nucleating particles (INPs) has been overlooked and this is one of the recent efforts to understand the intrinsic behaviour of this group of aerosol particles as INPs. This report has given an insight into the chemistry of CFA when immersed in water, which typifies the atmospheric process that these particles undergo in different cloud conditions. Although only a few samples are investigated here, the study has highlighted the deactivation of the ice-nucleating potential of CFA particles due to alterations in the chemical properties of the ash indicated by the two generation systems that they

employed in this study – dry and wet generation methods. The deactivation in the IN abilities of these particles observed for all the wet-generated CFA particles points to the possible chemical and physical changes that could occur in this scenario. Again, it is very striking to see that the IN ability of CFA can be compared to some mineral dust/mineral particles. This leaves an open question - what if the ice residue measurements attributed to mineral dust could also be contributed by CFA because of some similarities in their mineralogy? At the moment, I could not agree less with Grawe et al. that it is difficult to ascertain the atmospheric abundance as well as the transport of these particles in the atmosphere and hence, estimating its impact on clouds based on these new results will be a daunting task. On a general note, the manuscript is relatively well-composed, logically presented, and well-referenced – aside from the few major and minor comments that I have suggested below to enhance the quality of this work, I have no reservations in recommending this study for publication in ACP.

Minor Revisions

1. The differences between the efficiencies of CFA_dry and CFA_wet are really huge, some up to 6 orders of magnitude (Fig. 4) e.g. CFA4, do you think that only wet chemistry of the CFA can explain this difference? At least for CFA4, there are no needle-shaped particles observed. Are there other works that you can refer to that might give useful information to substantiate your hypothesis? Other workers in the ice nucleation community have seen these differences (though not this much). Please, can you explain more on these observations?

2. The authors should be consistent with the use of 'needle-shaped crystals or particles' rather than just needles unless clearly predefined. The ESEM images referred to in Figure S5 is of poor quality and that makes it highly difficult to appreciate the differences. The so-called 'needle-shaped' crystals are not well-shown in a convincing way. Please, do you have high quality or better ESEM images with high resolution that you can present to really buttress the findings that the report is making most references to? Please, could you clearly state how the particles from the wet aggregation method

were collected and treated before the ESEM imaging? One would expect a bit of aggregation of the particles if they were allowed to dry out from the droplets – otherwise, isolated particles will be seen. Figure S12 - further calls for a better interpretation of the morphology of the particles, Already, the scale shown is rather large for this type of study. Although there are crystalline pins present on the slide, there are also light-coloured regions, which may be CFA particles too. Please, could you throw more light on this?

3. For a better/easier readability, I would kindly recommend that Fig. 3 be modified following these suggestions: (1) show error bars in ns rather than temperature for CFA1-4 samples such that they can be easily compared with Garimella's work. Again, for the CFA3 panel, some selected data points can show the error bars to avoid obscuring its readability. (2) the legend should be kept together for an easy reference. (3) if possible, increase the data points size for Garimella's CFA and try to put a fit through the data points for an easy comparison.

4. The current title is a bit ambiguous for the results presented in this present study. The authors might consider focusing the title more on the 'reduction in the efficiency of CFA ice nucleation in the immersion freezing mode due to modifications/changes in the chemical properties'. OR another main idea that this work presents as a strong salient point is 'the difference between dry- and wet-generated CFA particles on their ice nucleation behaviour'. From this present investigation, it is not quite clear how the physical properties such as size, morphology and others change the ice nucleation behaviour rather attention should be focused on the unravelled chemical compositions and transformations. The authors might want to consider revising the title to carry the main idea of the article.

5. Figures (S6 – S9) - Please give more explanation in the plot's description, label the ordinates and abscissas correctly, and indicate the meaning of the legend codes (I.e. numbers in braces). Please, could you show the various classifications of CFA probably in a table format for readers to follow the discussion with ease? Indicate the equations

of the fit for Figure S11. Please remove the error numbers (±xxx) - they are irrelevant to the apparent BET surface areas as shown in Table S3. The errors are not realistic to the material rather to the fit/model performance hence, they are not needed (you may want to discuss with experts in this field). The adsorption model used in obtaining the specific surface area can vary considerably even with the same material. It really depends on how good the fit is and the number of points on the Isotherm considered.

Minor technical suggestions

Page 1, L4: This line should read '...physico-chemical properties of particles can influence...'.

Page 2, L1&2: Please replace "ice nucleating" with "ice-nucleating". This correction should be applied to such instances in the entire manuscript.

Page 2, L5: Please change 'there is ongoing discussion...' to 'there is an ongoing discussion...'.

Page 2, L6: Add Chen et al., 2018 to the reference.

Page 2, L9&10: Please restructure the sentence such that 'ice nucleation active' would be changed to something like 'act as ice nuclei'.

Page 2, L12: Please use past tense when describing past works in contexts like this one. e.g. Hoose and Möhler (2012) summarized...'. This has reoccurred in several places in the manuscript (next case is Page 2, L25 and so on). Please change "soot is a generally worse ice nucleus..." to "soot is generally a worse ice nucleus...".

Page 2, L17: Please give a better definition of bottom ash to distinguish the two forms of the ashes – one can also have bottom ash from the coal power plants, burning of agricultural fields, etc.

Page 2, L22: Please change '...gases from Coal Fly Ash.' to '...gases of Coal Fly Ash.'

Page 3, L1: I would omit the adjective 'perfectly'.

Page 3, L7: There are other works on ice nucleation of CFA captured by Umo et al., 2015 and Grawe et al. 2016 that preceded Havlicek's work. Please briefly mention them here or you can use an annotation like 'references therein' in an appropriate place.

Page 3, L13: Put a space before any unit e.g. -15 oC. Please correct this throughout the manuscript.

Page 3, L14-18: Please recast these statements – there are a bit confusing to me.

Page 3, L20: misspelt word "properties'.

Page 3, L26: This statement can read better as "Garimella (2016) investigated the freezing behaviour of four different CFA samples from the USA using the SPectrometer for Ice Nuclei (SPIN; Droplet Measurement Technologies, Inc.)".

Page 3, L28-29: Please could you indicate the relative humidity that 1% ice-activity was reported and same for Havlicek's study?

Page 3, L30: Please edit this line to read "…measurements of CFA by…".

Page 4, L1: Please restructure this line.

Page 4, L6: Please omit 'previously investigated' from this line.

Page 4, L12: These are very broad scientific questions. This part should be presented as hypotheses rather than as elaborate questions, which this project alone may not give all the answers.

Page 4, L14: Please specify the sort of deactivation referred to here. e.g. 'deactivation in the ice nucleation properties'

Page 4, L25: Please change 'consisted' to 'consists' and 'size selected' to 'size-selected'.
Page 4, L26: Please change 'multi Micro' to 'multi-micro'.

Page 5, L2-4: Can these pieces of information be obtained from the company? If not, ignore and recast the sentence in such a way to reflect the fact that your team was unable to get the information rather than connoting that it is not known.

Page 5, L8: 'Lime' – since this is a very generic term. Please distinguish between quicklime and slaked lime...I think 'CaO' is mostly referred to as quicklime. Stick to the right one all through the manuscript.

Page 5: I am not sure if footnotes are allowed in ACP – if not, integrate this information into section 2.1. Please check with the Editors.

Page 6, L11: Please change 'multiply charged' to 'multiply-charged'. Make this change in subsequent ones.

Page 6, L23: Please edit '0.5 wt% of CFA'. Did you mean that 0.5 g of CFA was dissolved in 100g of distilled water? Please check. Same for L33-Page 7, L1.

Page 7, L12: Please define 'fice' before use.

Page 7, L18: Please insert a comma after this statement 'The ice nucleation active surface site density'

Page 8, L20: Please correct to "...produced by a microfluidic device and subsequently arranged into..."

Page 8, L20: Please change 'pl' to 'pL'.

Page 8, L31: Please change "...the uncertainties of Vdrop, ..." to "...the uncertainties in the measurement of Vdrop,..."

Page 9, L4: Please change "...in the following, ..." to "...in the following sections, ....". Also, change 'analysis' to 'analyses'. Apply this to similar cases. E.g. Page 11, L17, etc.

Page 9, L5: Please recast the sentence to improve its readability. E.g. ...in the discussion of ... OR ...in discussing...

Page 9, L12-14: Please recast these statements.

Page 9, L24: Please correct 'CFA1 contains most Ca and S' to 'CFA1 has the highest concentration of Ca and S'. Please check the use of "most" and "least" in the entire manuscript. Sometimes I think you intended to use "highest" for "most" or "lowest" for "least".

Page 9, L28: Please correct "A more detailed' to 'A more-detailed'.

Page 10, L4: Include the company and country of the instrument in a parenthesis.

Page 10, L8-10: The information here is clear but check the use of tenses, verb agreements and possibly improve on the logical presentation of the observations here. E.g. The first sentence on L8 would read better as: 'CFA1 was the only sample that a clear difference was observed between the dry and wet particle generation methods'. However, check if this statement should come first.

Page 10, L31: Please include state symbols in this equation and all others in the manuscript where possible. E.g. CaO(s)...This will help readers to understand the chemistry better.

Page 11, L9: This line should read 'the occurrence of needle-shaped particles in wet-generated CFA1 could...'.

Page 11, L18: Please recast this sentence. Maybe, refer to the other samples as 'samples from the USA.

Page 12, L1: I am not sure what you referred to as "inhomogeneous ice nucleation properties". Please clarify.

Page 12, L8: Please include the temperature range e.g. at T< -xxx oC.

Page 13, Fig. 2: Please improve the colour-coding of the line type to distinguish them easily. A dashed line might be better.

Page 13, L12: Please change 'We compare the CFA results to cold stage measurements by compare' to 'We compared the CFA results to cold-stage measurements of Quartz by. . .'.

Page 14, L1-2: Please provide a reference to the statement about the ns values of dry-generated quartz particles.

Page 15, L16 – 19: Please rephrase these sentences for clarity. Were you referring to the 'particles soluble in the pure water'?

Page 16, L23: Please, are there previous references that you can point us to?

Page 16, L24: I would be a bit cautious in making the assertion that there is a good agreement for results from WISDOM at -35 oC because homogeneous freezing of pure water kicks just before this temperature as reported by Reicher et al., 2018 (Fig 5).

Page 16, L33: Please check this range '-15oC<T<-20oC'. . .did you want to write '-15 oC > T > -20 oC'? Please change 'levels off for. . .' to 'levels off from. . .'.

Page 17, L1-3: Check the statement - it seems there is no point stating this here if we cannot point to the data somewhere.

Page 18, L15: Please explain further what you mean by a layer (of what?). Please change 'In case of dry particle generation' to 'In the case of dry particle generation method'.

Page 20, L33: Delete the extra 'the'.

Page 21, L15: Please change 'l-1'to 'L-1' and in all other instances.

Page 22, L5,6,15,28: I would suggest the summary and conclusions section be revised to carry the main findings of the article without the questions. Some of the questions

are not well-answered by this single study as even argued by the authors (see Page 22, L9-14).

Page 23, L3: Please change '...decrease quickly in contact with water' to '...decrease quickly when in contact with water'.

Page 25, L8: Write out the 'US' in full.

S_Page 25, L20: Steenari et al., is listed in the references but not cited anywhere in the text...please check.

Associated References to this Review Report

Chen, J., Wu, Z., Augustin-Bauditz, S., Grawe, S., Hartmann, M., Pei, X., Liu, Z., Ji, D., and Wex, H.: Ice-nucleating particle concentrations unaffected by urban air pollution in Beijing, China, Atmos. Chem. Phys., 18, 3523-3539, https://doi.org/10.5194/acp-18-3523-2018, 2018.

Reicher, N., Segev, L., and Rudich, Y.: The Weizmann Supercooled Droplets Observation on a Microarray (WISDOM) and application for ambient dust, Atmos. Meas. Tech., 11, 233-248, https://doi.org/10.5194/amt-11-233-2018, 2018.

Grawe, S., Augustin-Bauditz, S., Hartmann, S., Hellner, L., Pettersson, J. B. C., Prager, A., Stratmann, F., and Wex, H.: The immersion freezing behavior of ash particles from wood and brown coal burning, Atmospheric Chemistry and Physics, 16, pp. 13 911–13 928, 2016.

Umo, N. S., Murray, B. J., Baeza-Romero, M. T., Jones, J. M., Lea-Langton, A. R., Malkin, T. L., O'Sullivan, D., Neve, L., Plane, J. M. C., and Williams, A.: Ice nucleation by combustion ash particles at conditions relevant to mixed-phase clouds, Atmospheric Chemistry and Physics, 15, pp. 5195–5210, 2015.

---

## Referee Comment (RC2) · Anonymous Referee #2 · 26 Jul 2018

This is a well-written paper concerning immersion freezing of water droplets triggered by coal fly ashes (CFA). This is a timely topic which fits very well into ACP. The authors compare samples from different sources concerning their ice nucleation activity. They use different set-ups (LACIS, WISDOM, SPIN, LINA) to do so. They correlate their findings with the physical-chemical properties of the particles. They conclude that $CaSO_4$ and $CaO$ are the crucial mineral components and that thus surface hydration of these fractions can have an important impact on the ice nucleation activity of CFA.

Particularly, the physical-chemical characterization of the ice nucleation particles (INPs) makes the importance of this paper. Therefore, the precise application of the

different methods in use is crucial. Therefore, I have listed here my concerns regarding the different techniques:

1. Alabama: I am not an expert in aerosol mass spectrometry. Therefore, I have no comments.

2. ESEM. The special resolution of the microscopic pictures is very low. I highly recommend transmission electron microscopy (TEM) pictures. This is of particular importance when investigating the spherical shaped combustion products, which are thought to origin from organics. TEM could provide the internal structure of the particles and will allow a correlation between structure and chemistry, which both have an impact on the ice nucleation (see e.g. Häusler et al. who have investigated typical constituents of soot and coal, i.e. graphenes, which have similar ns values like some coal fly ashes).

3. EDX. Is a valuable technique in order to gather the chemical composition of materials and is easily accessible in combination with SEM. However, at concentrations below 0.1% the signal to noise ratio of this technique in unsatisfying and the results are untrustworthy. I highly recommend using micro X-ray fluorescence analyses ($\mu$-XFA).

4. XRD. The powder diffractograms shown in the supplement are of excellent quality. Therefore, the authors can easily apply a Rietveld refinement of their data.

5. Bulk chemical composition analysis. Please specify how this was done.

6. DMPS. No comments.

7. Light microscopy. Eventually, polarization microscopy could help to differentiate the components of the particles (amorphous vs crystalline).

8. BET. Specific surface areas about 1 m2 g-1 are often not precisely accessible. Please, describe the detection limit of your instrument.

The authors should discuss in more detail the impact of internal structure, morphology and chemistryof the INPs on the ice nucleation activity. In particular, I miss a discussion

of the carbonaceous particles. This can easily been performed with the data at hand and with some modifications described above. Therefore, I rate this manuscript as "accepted, subject to minor revisions".

Reference

Häusler, H., Gebhardt, P., Iglesias, D., Rameshan, C., Marchesan, S., Eder, D., Grothe, H.: Ice Nucleation Activity of Graphene and Graphene Oxides, The Journal of Physical Chemistry C 122 (15), pp. 8182-8190, 2018. DOI: 10.1021/acs.jpcc.7b10675

---

## Author Comment (AC1) · 29 Aug 2018

**Answers to comments by anonymous Referee #1:**

We would like to thank Referee #1 for his/her helpful comments that certainly increase the readability of our manuscript and the overall quality of our study. In the following, we will address major and minor comments. For this, the referee comments will be given in green, our answers and adjustments to the manuscript in black. When referencing page and line numbers, we are always referring to the original versions of manuscript and SI.

The study reported by Sarah Grawe and her co-authors on linking the immersion freezing behaviour of CFA particles to the physicochemical properties of coal fly ash (CFA) is an interesting and timely investigation. For a long time, the study of CFA as ice-nucleating particles (INPs) has been overlooked and this is one of the recent efforts to understand the intrinsic behaviour of this group of aerosol particles as INPs. This report has given an insight into the chemistry of CFA when immersed in water, which typifies the atmospheric process that these particles undergo in different cloud conditions. Although only a few samples are investigated here, the study has highlighted the deactivation of the ice-nucleating potential of CFA particles due to alterations in the chemical properties of the ash indicated by the two generation systems that they employed in this study – dry and wet generation methods. The deactivation in the IN abilities of these particles observed for all the wet-generated CFA particles points to the possible chemical and physical changes that could occur in this scenario. Again, it is very striking to see that the IN ability of CFA can be compared to some mineral dust/mineral particles. This leaves an open question - what if the ice residue measurements attributed to mineral dust could also be contributed by CFA because of some similarities in their mineralogy? At the moment, I could not agree less with Grawe et al. that it is difficult to ascertain the atmospheric abundance as well as the transport of these particles in the atmosphere and hence, estimating its impact on clouds based on these new results will be a daunting task. On a general note, the manuscript is relatively well-composed, logically presented, and well-referenced – aside from the few major and minor comments that I have suggested below to enhance the quality of this work, I have no reservations in recommending this study for publication in ACP.

Minor Revisions

1. The differences between the efficiencies of CFA_dry and CFA_wet are really huge, some up to 6 orders of magnitude (Fig. 4) e.g. CFA4, do you think that only wet chemistry of the CFA can explain this difference? At least for CFA4, there are no needle-shaped particles observed. Are there other works that you can refer to that might give useful information to substantiate your hypothesis? Other workers in the ice nucleation community have seen these differences (though not this much). Please, can you explain more on these observations?

The largest difference in $n_s$ comparing dry- and wet generated particles (at -35 °C) was seen for CFA2 (~4 orders of magnitude), followed by CFA1 (~3 orders of magnitude), CFA4(~factor 180), and CFA3 (~1 order of magnitude). LACIS and LINA, where we assume you found the six orders of magnitude difference for CFA4, cannot be compared because there is no temperature overlap. It is indeed interesting that the largest difference was not observed for CFA1, which presumably contains most soluble anhydrite, but for CFA2. However, when considering the error bars, which are quite large for wet-generated particles (for explanation see P15, L9-11), it becomes obvious that we should be cautious when comparing the degree of reduction in $n_s$ for the different samples. It must be noted that even if the needle-shaped particles were only observed for CFA1, we also found hydration products in wet-generated particles of the other three samples (see Sec. S1 and S3).

We agree that it would help to reference previous studies to substantiate our hypothesis of hydration leading to a reduction in $n_s$ when switching from dry to wet particle generation. But to our knowledge there are no published immersion freezing measurements of dry- and wet-generated particles of a hydratable substance with a single particle instrument such as LACIS. The studies you probably refer to (e.g., Hiranuma et al, 2015; Emersic et al., 2015) compare $n_s$ values derived from cold stage methods and dry-dispersed particle measurements, not dry- and wet-dispersed particle measurements with one instrument. Hence, hypotheses given as possible explanations for the observed discrepancy between suspension methods and dry-dispersed particle measurements given in Hiranuma et al. (2015) and Emersic et al. (2015) for the most part do not apply to our measurements with dry- and wet-generated size-selected particles.

To summarize: Yes, we do assume that aqueous chemistry causes the observed reduction in $n_s$. We slightly adapted the paragraph dealing with this topic (P16L6-11): "A decrease in immersion freezing efficiency from dry to wet particle generation was already reported for CFA and coal bottom ash in Grawe et al. (2016). A possible explanation for the observed discrepancy was presented following previous investigations of Hiranuma et al. (2015), who conducted immersion freezing measurements with both dry-dispersed mineral dust and mineral dust suspensions. There, it was hypothesized that the increased time that the particles spend in contact with water leads to a change in chemical particle properties. For our previous study (Grawe et al., 2016), it was not possible to identify relevant processes because information on the chemical composition of 300 nm particles was missing. In the framework of the present study, differences in chemical composition of dry- and wet-generated CFA particles were identified (see Sec. 2.4 and S1), and will be discussed in relation with the immersion freezing results in Sec. 3.2.2.".

> 2. The authors should be consistent with the use of 'needle-shaped crystals or particles' rather than just needles unless clearly predefined. The ESEM images referred to in Figure S5 is of poor quality and that makes it highly difficult to appreciate the differences. The so-called 'needle-shaped' crystals are not well-shown in a convincing way. Please, do you have high quality or better ESEM images with high resolution that you can present to really buttress the findings that the report is making most references to? Please, could you clearly state how the particles from the wet aggregation method were collected and treated before the ESEM imaging? One would expect a bit of aggregation of the particles if they were allowed to dry out from the droplets – otherwise, isolated particles will be seen. Figure S12 - further calls for a better interpretation of the morphology of the particles, Already, the scale shown is rather large for this type of study. Although there are crystalline pins present on the slide, there are also light-coloured regions, which may be CFA particles too. Please, could you throw more light on this?

We changed the expression to "needle-shaped particles" in all instances.

The wet-generated particles were collected behind the DMA as shown in Fig. 1. with the multi-MINI. There was no treatment to the loaded substrates before ESEM measurements. This was added to the manuscript in Sec. 2.4.1. Please be aware that the particles were dried already before coming into contact with the substrate, they do not hit the substrate as droplets and dry there.

At the moment it is not possible for us to provide higher resolution ESEM images as the instruments are currently upgraded. An ESEM image was added in the SI (Fig. S6) on which the needles are clearly visible.

The images from the optical microscope (now Fig. S13) are only shown to stress that the needle-shaped particles precipitate in the suspension in case of CFA1 which explains the large scale of the images. We do not see any need to include new images with higher magnification. We added the following paragraph to explain Fig. S13 in more detail: "Images of liquid CFA suspension droplets were taken with a digital camera coupled to an optical microscope (Primovert, Carl Zeiss Microscopy

GmbH, Jena, Germany). The magnification is 200x and unpolarized light was used. The suspensions were prepared in the same way as for the LACIS measurements and pipetted onto a glass microscope slide. A second slide was put on top of the liquid droplet to increase the amount of particles in focus and to avoid evaporation during examination. Figure S13 a shows that needle-shaped particles are present in the aqueous environment of the CFA1 suspension, suggesting that they precipitate in the suspension and are not or only weakly water-soluble. The needle-shaped particles are several tens of microns long. In addition to the needle-shaped particles, smaller spherical and irregularly shaped particles can be seen. Droplets from the CFA2, CFA3, and CFA4 suspensions do not contain needle-shaped particles, only irregular and spherical particles. Generally, the number of irregularly shaped particles visible in Fig. S13 is much higher than the number of spherical particles for all samples. Coagulation of particles can be observed to some extent for all samples and might affect the surface area available for triggering immersion freezing in the cold stage experiments as described by Emersic et al. (2015)."

3. For a better/easier readability, I would kindly recommend that Fig. 3 be modified following these suggestions: (1) show error bars in ns rather than temperature for CFA1-4 samples such that they can be easily compared with Garimella's work. Again, for the CFA3 panel, some selected data points can show the error bars to avoid obscuring its readability. (2) the legend should be kept together for an easy reference. (3) if possible, increase the data points size for Garimella's CFA and try to put a fit through the data points for an easy comparison.

The uncertainty in the SPIN $AF$ is 14 % due to 10 % uncertainty of OPC and CPC. This value was considered for estimating the uncertainty in $n_s$ ( $n_s = -\ln(AF-1)/A_p$) which was added as vertical error bars in all panels. For CFA1 and CFA3, we only show error bars in $n_s$ and omit the horizontal error bars for greater clarity. Explanatory sentences were added in Sec. 2.3.2 and 3.1.1. Note that the vertical error bars for Garimella (2016) are derived from the uncertainty of a machine learning approach that was used to determine $AF$ and hence differ substantially from ours. The data point size for data from Garimella (2016) was increased and an exponential fit was added. The legend was changed as suggested. Error bars for the SPIN $AF$ were also added in Fig. C1.

Note that the upper left panel now includes more data points of measurements with CFA1 than in the previous version of the manuscript. In the previous version, there was an error in the generation of Fig. 3 which was now discovered and corrected. The main message, i.e., that the CFA investigated by Garimella (2016) is significantly different from CFA1 in terms of its immersion freezing behavior, is not changed by this.

4. The current title is a bit ambiguous for the results presented in this present study. The authors might consider focusing the title more on the 'reduction in the efficiency of CFA ice nucleation in the immersion freezing mode due to modifications/changes in the chemical properties'. OR another main idea that this work presents as a strong salient point is 'the difference between dry- and wet-generated CFA particles on their ice nucleation behaviour'. From this present investigation, it is not quite clear how the physical properties such as size, morphology and others change the ice nucleation behaviour rather attention should be focused on the unravelled chemical compositions and transformations. The authors might want to consider revising the title to carry the main idea of the article.

We feel that the given suggestions do not reach far enough as physical properties were also investigated and discussed in connection with the immersion freezing results. On P16L24-26, e.g., we discuss the good agreement between LACIS wet and WISDOM for CFA1 which shows that the immersion freezing efficiency of this sample scales with the surface area of the particles. In addition we elaborately discuss changes in morphology (occurrence of needle-shaped particles) and crystallography (anhydrite-gypsum and quicklime-calcite conversions) and their effects on the

immersion freezing behavior. Hence, we would like to stick with the current title unless a change is absolutely requested for publication.

5. Figures (S6 – S9) - Please give more explanation in the plot's description, label the ordinates and abscissas correctly, and indicate the meaning of the legend codes (I.e. numbers in braces). Please, could you show the various classifications of CFA probably in a table format for readers to follow the discussion with ease? Indicate the equations of the fit for Figure S11. Please remove the error numbers ( ± xxx) - they are irrelevant to the apparent BET surface areas as shown in Table S3. The errors are not realistic to the material rather to the fit/model performance hence, they are not needed (you may want to discuss with experts in this field). The adsorption model used in obtaining the specific surface area can vary considerably even with the same material. It really depends on how good the fit is and the number of points on the Isotherm considered.

Figures S7-S10 (former Fig. S6-S9) and their captions were modified according to your suggestions. We included a Table (S2) showing the identified major (> 5%, checkmarks) and minor phases (< 5%, checkmarks in parentheses). We decided not to include percentages in Table S2 because amounts of identified phases can vary significantly between measurements (see P11L5-6 of the SI).

Instead of including the normal distribution fit functions in Fig. S12 (formerly S11), we present a Table (S3) showing μ and σ of both modes for each sample.

The errors of the BET measurements shown in Table S5 (former Table S3) are the standard deviation of values derived from three measurements with $R^2$=0.999, i.e., very accurate fits to the data. We used 11-point BET. We do not quite understand why our error bars should not be meaningful but are open to suggestions for improvement. For now, nothing was changed concerning the error bars, but we included some more information, i.e., the accuracy of the fits, the number of points considered, and the detection limit of the instrument (0.01 $m^2$ $g^{-1}$) in Sec. S8.

Minor technical suggestions

Page 1, L4: This line should read '. . .physico-chemical properties of particles can influence. . .'.
Changed.

Page 2, L1&2: Please replace "ice nucleating" with "ice-nucleating". This correction should be applied to such instances in the entire manuscript.
Done.

Page 2, L5: Please change 'there is ongoing discussion...' to 'there is an ongoing discussion...'.
Done.

Page 2, L6: Add Chen et al., 2018 to the reference.
Done.

Page 2, L9&10: Please restructure the sentence such that 'ice nucleation active' would be changed to something like 'act as ice nuclei'.
Done.

Page 2, L12: Please use past tense when describing past works in contexts like this one. e.g. Hoose and Möhler (2012) summarized. . .'. This has reoccurred in several places in the manuscript (next case is Page 2, L25 and so on). Please change "soot is a generally worse ice nucleus..." to "soot is generally a worse ice nucleus. . .".

Tenses were adjusted to past where necessary throughout the manuscript.
We would like to keep the wording of the cited phrase because it is a direct quote taken from Hoose and Möhler (2012).

Page 2, L17: Please give a better definition of bottom ash to distinguish the two forms of the ashes – one can also have bottom ash from the coal power plants, burning of agricultural fields, etc.
We now terminate the sentence after "bottom ash" and included the following: "The latter is defined as the fraction that remains in the power plant, fireplace, or on the ground after a wildfire and can be emitted due to wind erosion."

Page 2, L22: Please change '. . .gases from Coal Fly Ash.' to '. . .gases of Coal Fly Ash.'
Changed.

Page 3, L1: I would omit the adjective 'perfectly'.
Deleted.

Page 3, L7: There are other works on ice nucleation of CFA captured by Umo et al., 2015 and Grawe et al. 2016 that preceded Havlicek's work. Please briefly mention them here or you can use an annotation like 'references therein' in an appropriate place.
We incorporated the earlier studies in the following way: "Concerning the ice nucleation activity of ash particles, only few studies have been published so far. Early investigations of aerosol from coal-fired power plant plumes were contradictory as to whether the particles are able to act as INPs (Parungo et al., 1978) or not (Schnell et al. 1976). More recent studies (Havlíček et al., 1993; Umo et al., 2015; Garimella, 2016; Grawe et al., 2016) agreed that ash particles indeed trigger heterogeneous ice nucleation."

Page 3, L13: Put a space before any unit e.g. -15 °C. Please correct this throughout the manuscript.
Done.

Page 3, L14-18: Please recast these statements – there are a bit confusing to me.
We rephrased the sentences in the following way: "The water-soluble components were separated from all samples and ice nucleation experiments were carried out with the original samples, the water-insoluble components, and the water-soluble components. Immersion freezing was found to be less efficient than deposition nucleation in all cases. The water-insoluble components were up to three orders of magnitude less efficient in the deposition mode than the untreated samples. However, when the water-soluble components alone were investigated, they showed surprisingly low efficiency. This finding illustrates the complex interplay of physico-chemical particle properties and freezing behavior, as the water-soluble components increased the ice nucleation efficiency only when associated with the CFA particles."

Page 3, L20: misspelt word "properties".
Corrected.

Page 3, L26: This statement can read better as "Garimella (2016) investigated the freezing behaviour of four different CFA samples from the USA using the Spectrometer for Ice Nuclei (SPIN; Droplet Measurement Technologies, Inc.)".
Changed.

Page 3, L28-29: Please could you indicate the relative humidity that 1% ice-activity was reported and same for Havlicek's study?
No relative humidity is given for the deposition nucleation experiments by Havlíček et al. (1993). Garimella (2016) report $AF$=1% for $T$<-30 °C at 1.25 < $S_{ice}$< 1.4, which is now included in the manuscript.

Page 3, L30: Please edit this line to read ". . .measurements of CFA by. . ..".
Done.

Page 4, L1: Please restructure this line.
Done. It now reads: "Previously (Grawe et al., 2016), we investigated the freezing behavior of wood bottom ash, coal bottom ash, and CFA".

Page 4, L6: Please omit 'previously investigated' from this line.
Removed.

Page 4, L12: These are very broad scientific questions. This part should be presented as hypotheses rather than as elaborate questions, which this project alone may not give all the answers.
We do not really see how presenting the questions as hypotheses would make a difference here. We clearly say that we "aim" at answering the questions, which worked out for most of them. We changed the sentence following the questions to: "Four CFA samples from German power plants were investigated as immersion INPs in an attempt to answer these questions."

Page 4, L14: Please specify the sort of deactivation referred to here. e.g. 'deactivation in the ice nucleation properties'
Changed as suggested.

Page 4, L25: Please change 'consisted' to 'consists' and 'size selected' to 'size-selected'.
Changed.

Page 4, L26: Please change 'multi Micro' to 'multi-micro'.
Changed.

Page 5, L2-4: Can these pieces of information be obtained from the company? If not, ignore and recast the sentence in such a way to reflect the fact that your team was unable to get the information rather than connoting that it is not known.
In the case of the Lippendorf sample it was not possible to obtain the information. The other samples were given to us through an intermediary and we do not know from which power plants they originate. Hence, also technical details are unknown to us. We added the following sentence: "This technical information could either not be obtained, or is unknown to us because the sample origin must remain anonymous."

Page 5, L8: 'Lime' – since this is a very generic term. Please distinguish between quicklime and slaked lime. . .I think 'CaO' is mostly referred to as quicklime. Stick to the right one all through the manuscript.
Thanks for pointing this out. Indeed, quicklime is the correct term in our case. We now use this term throughout the manuscript.

Page 5: I am not sure if footnotes are allowed in ACP – if not, integrate this information into section 2.1. Please check with the Editors.

The ACP website states that footnotes "should be avoided, as they tend to disrupt the flow of the text. If absolutely necessary, they should be numbered consecutively", hence they are not strictly forbidden. We would like to keep the footnote because we feel that the working principle of an electrostatic precipitator should be explained at this point. Including this paragraph in the text would probably disrupt the flow more than the footnote.

Page 6, L11: Please change 'multiply charged' to 'multiply-charged'. Make this change in subsequent ones.

Changed.

Page 6, L23: Please edit '0.5 wt% of CFA'. Did you mean that 0.5 g of CFA was dissolved in 100g of distilled water? Please check. Same for L33-Page 7, L1.

Thanks for pointing this out. Indeed, we mixed 0.5 g CFA with 100 mL water. This was changed throughout the manuscript.

Page 7, L12: Please define 'fice' before use.

The frozen fraction is first mentioned on Page 6, L12. We added "frozen fraction ($f_{ice}$, number of frozen hydrometeors divided by total number of hydrometeors)" there.

Page 7, L18: Please insert a comma after this statement 'The ice nucleation active surface site density'

We are quite sure that no comma is needed. Nothing was changed.

Page 8, L20: Please correct to ". . .produced by a microfluidic device and subsequently arranged into. . ."

Corrected.

Page 8, L20: Please change 'pl' to 'pL'.

Changed, also in all other instances.

Page 8, L31: Please change ". . .the uncertainties of Vdrop, . . ." to ". . .the uncertainties in the measurement of Vdrop,. . ."

Done.

Page 9, L4: Please change ". . .in the following, . . ." to ". . .in the following sections, . . .". Also, change 'analysis' to 'analyses'. Apply this to similar cases. E.g. Page 11, L17, etc.

Done.

Page 9, L5: Please recast the sentence to improve its readability. E.g. . . .in the discussion of . . . OR . . .in discussing. . .

Done.

Page 9, L12-14: Please recast these statements.

The sentences were changed: "In the following, the particles pass two subsequent detection lasers (wavelength of 405 nm). Information about the time-of-flight between the detection lasers is needed to trigger the ablation laser. In addition, the time-of-flight can be used to calculate the particle

vaccuum aerodynamic diameter for particles in a size range between ~200 and 2500 nm. The ablation laser, a ..."

Page 9, L24: Please correct 'CFA1 contains most Ca and S' to 'CFA1 has the highest concentration of Ca and S'. Please check the use of "most" and "least" in the entire manuscript. Sometimes I think you intended to use "highest" for "most" or "lowest" for "least".
Changed throughout the manuscript.

Page 9, L28: Please correct "A more detailed' to 'A more-detailed'.
Corrected, also in the other instances.

Page 10, L4: Include the company and country of the instrument in a parenthesis.
We added "(FEI Company, Hillsboro, OR, USA)" to the manuscript.

Page 10, L8-10: The information here is clear but check the use of tenses, verb agreements and possibly improve on the logical presentation of the observations here. E.g. The first sentence on L8 would read better as: 'CFA1 was the only sample that a clear difference was observed between the dry and wet particle generation methods'. However, check if this statement should come first.
We recast the sentences and hope that readability is improved: "The ESEM images (see Fig. S5 and S6) show that CFA1 is special in terms of particle morphology. Dry-generated CFA1 particles consist of irregularly shaped agglomerates of small spherules, which were not observed to this extent for the other samples. Wet-generated CFA1 particles appear to be an external mixture of spherules and needle-shaped crystals. CFA1 is the only sample for which needles were observed in connection with wet particle generation, and also the only sample for which a clear difference in morphology was observed between the dry and wet particle generation methods."

Page 10, L31: Please include state symbols in this equation and all others in the manuscript where possible. E.g. CaO(s). . .This will help readers to understand the chemistry better.
State symbols were added.

Page 11, L9: This line should read 'the occurrence of needle-shaped particles in wet-generated CFA1 could. . .'.
Changed.

Page 11, L18: Please recast this sentence. Maybe, refer to the other samples as 'samples from the USA.
The sentence was changed to: "Figure 3 shows SPIN results of measurements with the German CFA samples and four CFA samples from the USA (Garimella, 2016)." We now refer to "samples from the USA" instead of "U.S. American samples" in all cases.

Page 12, L1: I am not sure what you referred to as "inhomogeneous ice nucleation properties". Please clarify.
We omitted the cited expression and changed the passage in the following way: "The broad temperature range, in which the increase in $n_s$ is observed, hints at a variety of different types of ice nucleation active sites at the surface of the CFA particles. In case of uniform ice nucleation properties, a steep increase would be observed."

Page 12, L8: Please include the temperature range e.g. at T< -xxx oC.
Done.

Page 13, Fig. 2: Please improve the colour-coding of the line type to distinguish them easily. A dashed line might be better.
Both lines are now thicker, the lower is dashed and black. They are now easily distinguishable.

Page 13, L12: Please change 'We compare the CFA results to cold stage measurements by compare' to 'We compared the CFA results to cold-stage measurements of Quartz by. . .'.
Done.

Page 14, L1-2: Please provide a reference to the statement about the ns values of dry-generated quartz particles.
Unfortunately, the LACIS quartz measurements were test measurements and never published, i.e., there is no reference. We decided on not mentioning the LACIS quartz data as we do not wish to publish them here. The respective sentence was adapted in the following way: "We compared the CFA results to cold stage measurements of quartz by Atkinson et al. (2013) here because this dataset spans the relevant temperature range and because there is a lack of immersion freezing results of dry-generated quartz in the literature."

Page 15, L16 – 19: Please rephrase these sentences for clarity. Were you referring to the 'particles soluble in the pure water'?
No, we are referring to purely water-soluble particles, i.e., particles which do not contain water-insoluble material. This was added to the manuscript. Also, we reference Sec. S5, which contains more information concerning this matter.

Page 16, L23: Please, are there previous references that you can point us to?
Are you referring to previous intercomparisons of LINA and WISDOM? As both are relatively new instruments, this is the first intercomparison between the two. WISDOM and BINARY (Budke and Koop, 2015), which is similar to LINA, compared well for NX illite (Reicher et al., 2018). But CFA is a completely different substance and hence we would like to avoid referring to a previous intercomparision with a mineral dust suspension. Nothing was changed.

Page 16, L24: I would be a bit cautious in making the assertion that there is a good agreement for results from WISDOM at -35 °C because homogeneous freezing of pure water kicks just before this temperature as reported by Reicher et al., 2018 (Fig 5).
Indeed, first freezing of pure water droplets is observed at -35 °C with WISDOM. However, the WISDOM measurements were conducted for $T \geq$ -34.5 °C and so we are quite confident that the major contribution to freezing at this temperature is due to heterogeneous nucleation. Of course it cannot be ruled out that a few droplets freeze due to homogeneous nucleation at -34.5 °C. We added the following sentence to the manuscript: "At this temperature, the contribution of homogeneous nucleation is still minor in WISDOM measurements and hence we conclude that the major contribution to the observed freezing behavior is due to immersion freezing triggered by CFA particles."

Page 16, L33: Please check this range '-15 °C<T<-20 °C'. . .did you want to write '-15 °C > T > -20 °C'? Please change 'levels off for. . .' to 'levels off from. . .'.
Corrected.

Page 17, L1-3: Check the statement - it seems there is no point stating this here if we cannot point to the data somewhere.

This statement was included because there are no published intercomparisons between LINA and the µL-NIPI. We would like to avoid showing this comparison because a bottom ash sample was investigated on both instruments which does not fit within the scope of our paper. The comparison is shown below, but nothing was changed in the manuscript.

[Figure]

Page 18, L15: Please explain further what you mean by a layer (of what?). Please change 'In case of dry particle generation' to 'In the case of dry particle generation method'.

We do not know for sure what the layer is made of. We changed the sentence to: "The fact that the other samples also contain significant amounts of quartz, both in 300 nm particles and in bulk, and, nevertheless, feature a much lower efficiency, supports the hypothesis of the particles being coated by a layer which suppresses the ice nucleation efficiency of the quartz."
The beginning of the next sentence was changed as suggested.

Page 20, L33: Delete the extra 'the'.
Done.

Page 21, L15: Please change 'l-1'to 'L-1' and in all other instances.
Changed.

Page 22, L5,6,15,28: I would suggest the summary and conclusions section be revised to carry the main findings of the article without the questions. Some of the questions are not well-answered by this single study as even argued by the authors (see Page 22, L9-14).

We would like to keep the questions as they nicely frame our manuscript. In the beginning we state that we aim at answering the questions, so it is not necessarily expected that we also find answers to all of them. We exchanged "We can now give answers to the following questions from the introduction" with "In light of our new findings, we now revisit the questions from the introduction".

Page 23, L3: Please change '. . .decrease quickly in contact with water' to '. . .decrease quickly when in contact with water'.
Changed.

Page 25, L8: Write out the 'US' in full.
Done.

Page 25, L20: Steenari et al., is listed in the references but not cited anywhere in the text. . .please check.
Removed.

Associated References to this Review Report

Chen, J., Wu, Z., Augustin-Bauditz, S., Grawe, S., Hartmann, M., Pei, X., Liu, Z., Ji, D., and Wex, H.: Ice-nucleating particle concentrations unaffected by urban air pollution in Beijing, China, Atmos. Chem. Phys., 18, 3523-3539, https://doi.org/10.5194/acp-18-3523-2018, 2018.

Reicher, N., Segev, L., and Rudich, Y.: The Weizmann Supercooled Droplets Observation on a Microarray (WISDOM) and application for ambient dust, Atmos. Meas. Tech., 11, 233-248, https://doi.org/10.5194/amt-11-233-2018, 2018.

Grawe, S., Augustin-Bauditz, S., Hartmann, S., Hellner, L., Pettersson, J. B. C., Prager, A., Stratmann, F., and Wex, H.: The immersion freezing behavior of ash particles from wood and brown coal burning, Atmospheric Chemistry and Physics, 16, pp. 13 911–13928, 2016.

Umo, N. S., Murray, B. J., Baeza-Romero, M. T., Jones, J. M., Lea-Langton, A. R., Malkin, T. L., O'Sullivan, D., Neve, L., Plane, J. M. C., and Williams, A.: Ice nucleation by combustion ash particles at conditions relevant to mixed-phase clouds, Atmospheric Chemistry and Physics, 15, pp. 5195–5210, 2015.

References mentioned in our answers (which are not part of the originally submitted manuscript):

Emersic, C., Connolly, P. J., Boult, S., Campana, M., and Li, Z.: Investigating the discrepancy between wet-suspension- and dry dispersion-derived ice nucleation efficiency of mineral particles, Atmospheric Chemistry and Physics, 15, pp. 11 311–11 326, 2015.

Hiranuma, N., Augustin-Bauditz, S., Bingemer, H., Budke, C., Curtius, J., Danielczok, A., Diehl, K., Dreischmeier, K., Ebert, M., Frank, F., Hoffmann, N., Kandler, K., Kiselev, A., Koop, T., Leisner, T., Möhler, O., Nillius, B., Peckhaus, A., Rose, D., Weinbruch, S., Wex, H., Boose, Y., DeMott, P. J., Hader, J. D., Hill, T. C. J., Kanji, Z. A., Kulkarni, G., Levin, E. J. T., McCluskey, C. S., Murakami, M., Murray, B. J., Niedermeier, D., Petters, M. D., O'Sullivan, D., Saito, A., Schill, G. P., Tajiri, T., Tolbert, M. A., Welti, A., Whale, T. F., Wright, T. P., and Yamashita, K.: A comprehensive laboratory study on the immersion freezing behavior of illite NX particles: A comparison of 17 ice nucleation measurement techniques, Atmospheric Chemistry and Physics, 15, pp. 2489–2518, 2015.

Schnell, R. C., Valin, C. C. V., and Pueschel, R. F.: Atmospheric ice nuclei: No detectable effect from a coal-fired powerpower plume, Geophysical Research Letters, Vol. 3, No. 11, pp. 657–660, 1976.

---

## Author Comment (AC2) · 29 Aug 2018

**Answers to comments by Anonymous Referee #2:**

We would like to thank Referee #2 for his/her helpful comments that certainly increase the quality of our manuscript. In the following, the referee comments will be given in green, our answers and adjustments to the manuscript in black. When referencing page and line numbers, we are always referring to the original versions of manuscript and SI.

This is a well-written paper concerning immersion freezing of water droplets triggered by coal fly ashes (CFA). This is a timely topic which fits very well into ACP. The authors compare samples from different sources concerning their ice nucleation activity. They use different set-ups (LACIS, WISDOM, SPIN, LINA) to do so. They correlate their findings with the physical-chemical properties of the particles. They conclude that CaSO4 and CaO are the crucial mineral components and that thus surface hydration of these fractions can have an important impact on the ice nucleation activity of CFA.

Particularly, the physical-chemical characterization of the ice nucleation particles (INPs) makes the importance of this paper. Therefore, the precise application of the different methods in use is crucial. Therefore, I have listed here my concerns regarding the different techniques:

1. Alabama: I am not an expert in aerosol mass spectrometry. Therefore, I have no comments.

2. ESEM. The special resolution of the microscopic pictures is very low. I highly recommend transmission electron microscopy (TEM) pictures. This is of particular importance when investigating the spherical shaped combustion products, which are thought to origin from organics. TEM could provide the internal structure of the particles and will allow a correlation between structure and chemistry, which both have an impact on the ice nucleation (see e.g. Häusler et al. who have investigated typical constituents of soot and coal, i.e. graphenes, which have similar ns values like some coal fly ashes).

As stated on P2L13-15 of the main text, CFA is largely composed of non-combustible constituents in the fuel, i.e., mineral inclusions, and not organics. This is a fundamental difference between ash and carbonaceous particles. Graphene might be a typical constituent of coal, but it is not common in CFA. The ESEM instrument is currently upgraded and not available for further analysis. We agree that TEM analysis would be interesting, however, we would have to resample because the used boron substrates are not suitable for TEM. Resampling and analyzing the eight different particle types (CFA1-CFA4, dry and wet, respectively) would be a time-consuming process. Considering that the focus of the paper is on the investigation of the immersion freezing behavior of CFA, and considering that new images would not change the main message of the study, performing these additional measurements would be beyond the scope of this work. We now cite Gieré et al. (2003) who performed TEM with CFA in the SI (Sec. S2): "Gieré et al. (2003) who performed transmission electron microscopy of class F CFA particles found both, smooth spherical particles and irregularly shaped particles in the size range of several hundred nanometers. The irregularly shaped particles were made up of crushed glass, or glassy spheres with small crystals attached to their surface which concealed the spherical shape.".

3. EDX. Is a valuable technique in order to gather the chemical composition of materials and is easily accessible in combination with SEM. However, at concentrations below 0.1% the signal to noise ratio of this technique in unsatisfying and the results are untrustworthy. I highly recommend using micro X-ray fluorescence analyses (μ-XFA).

We assume that the absence of, e.g., Ba, Ti, Sr, and Pb in the EDX results can be largely attributed to the weak statistics (~20 particles were investigated for each sample) and not to the detection limit of the EDX detector. Weinbruch et al. (2010) detected Ti in CFA particles using EDX. As stated in the SI (P8L21-22), our substrates were unfortunately not loaded ideally which is why resampling would be necessary to provide more information. EDX was included in addition to ALABAMA because, as you correctly stated, it is easily accessible in combination with the ESEM.

We are quite certain that μ-XRF would not yield satisfactory results when applied to 300 nm particles. Also, further information concerning trace metal occurrence in single 300 nm particles, which we already have from ALABAMA, is not needed from our point of view.

4. XRD. The powder diffractograms shown in the supplement are of excellent quality. Therefore, the authors can easily apply a Rietveld refinement of their data.

As stated in the main text (P10L22-23) "quantitative phase identification was done by Rietveld refinement using reference patterns from the Crystallography Open Database (Gražulis et al., 2009)". We now include a table (new Table S2) showing the identified phases for each sample in the SI.

5. Bulk chemical composition analysis. Please specify how this was done.

Please refer to the main text (P11L5-7).

6. DMPS. No comments.

7. Light microscopy. Eventually, polarization microscopy could help to differentiate the components of the particles (amorphous vs crystalline).

The light microscopy images were included to show that needle-shaped particles exclusively occur in the aqueous environment of the CFA1 suspension. Further polarization microscopy images would most certainly show the same as the XRD results, i.e., that CFA3 contains most amorphous material. A more conclusive statement about the amorphous fraction is not expected to change the interpretation of the immersion freezing results, which is why nothing was changed in this regard. We added the following paragraph to Sec. S6 for further explanation of the light microscopy images: "Images of liquid CFA suspension droplets were taken with a digital camera coupled to an optical microscope (Primovert, Carl Zeiss Microscopy GmbH, Jena, Germany). The magnification is 200x and unpolarized light was used. The suspensions were prepared in the same way as for the LACIS measurements and pipetted onto a glass microscope slide. A second slide was put on top of the liquid droplet to increase the amount of particles in focus and to avoid evaporation. Figure S13 a shows that needle-shaped particles are present in the aqueous environment of the CFA1 suspension, suggesting that they precipitate in the suspension and are not or only weakly water-soluble. The needle-shaped particles are several tens of microns long. In addition to the needle-shaped particles, smaller spherical and irregularly shaped particles can be seen. Droplets from the CFA2, CFA3, and CFA4 suspensions do not contain needle-shaped particles, only irregular and spherical particles. Generally, the number of irregularly shaped particles visible in Fig. S13 is much higher than the number of spherical particles for all samples. Coagulation of particles can be observed to some extent for all samples and might affect the surface area available for triggering immersion freezing in the cold stage experiments as described by Emersic et al. (2015).".

8. BET. Specific surface areas about 1 m2 g-1 are often not precisely accessible. Please, describe the detection limit of your instrument.

The detection limit of the instrument (Nova 2200e, Quantachrome Instruments, Boynton Beach, FL, USA) is 0.01 $m^2 g^{-1}$. This was added in Sec. S8.

The authors should discuss in more detail the impact of internal structure, morphology and chemistry of the INPs on the ice nucleation activity. In particular, I miss a discussion of the carbonaceous particles. This can easily been performed with the data at hand and with some modifications described above. Therefore, I rate this manuscript as "accepted, subject to minor revisions".

Concerning the internal structure of the particles, we only have very limited knowledge from the BET measurements, i.e., that CFA4 classifies as porous whereas the other samples are non-porous. Hence, it is difficult for us to discuss the ice nucleation efficiency of the samples in connection with their internal structure. From our point of view, the discussion of morphology (see Sec. S2, S6), crystallography (Sec. S3), and especially chemical composition (Sec. S1, S4) and possible links between immersion freezing behavior and physico-chemical particle properties (see Sec. 3.1.2, 3.2.2, 3.2.3, and especially 3.3) is very detailed in our study already.

As stated in the introduction of the main text (P2L13), CFA only contains a limited amount of carbon. This is due to the very efficient and almost complete combustion of pulverized coal in power plants. Loss on ignition values, which are now included in the SI and shortly discussed in the main text, clearly show that carbon is a minor component of the CFA samples. Only the freezing behavior of CFA4, which contains 8 ± 5 % of unburnt fuel might be influenced by the occurrence of carbonaceous particles. The following sentences were added to the main text (Sec. 3.2.1): "According to LOI measurements (see Sec. S4), CFA4 contains the highest amount of unburnt fuel, which is presumably made up of carbonaceous particles. The low immersion freezing efficiency of CFA4 in the investigated temperature range could hence be related to the occurrence of carbonaceous particles, which have previously been found to be inefficient at nucleating ice in the immersion mode (e.g., Chen et al., 2018)". The following sentences were added to the SI (Sec. S4): "In addition to the bulk chemical composition analysis, Loss On Ignition (LOI) values were determined. The LOI value is a measure of the amount of unburnt fuel, presumably carbonaceous particles, in the CFA samples and hence useful to assess the completeness of combustion in the power plant. The LOI values of the four CFA samples are -0.8 ± 5 % for CFA1, 0.2 ± 5 % for CFA2, 0.8 ± 5 % for CFA3, and 8.1 ± 5 % for CFA4, i.e., apparently only CFA4 still contains a relevant amount of unburnt fuel after combustion in the power plant. Particles with high C content tend to form irregular structures because of enhanced aggregation (Hiranuma et al., 2008). The specific surface area of CFA4, which is more than one order of magnitude higher than that of the other samples (see Sec. S8), could hence be in line with the comparably large LOI value. The fact that CFA4 has the lowest immersion freezing efficiency of all samples in the LINA experiments might be related to the amount of unburnt fuel in this sample. Carbonaceous particles, such as soot, have previously been shown to possess limited ice nucleation efficiency in the immersion mode (e.g., Chen et al., 2018).".

Reference
Häusler, H., Gebhardt, P., Iglesias, D., Rameshan, C., Marchesan, S., Eder, D., Grothe, H.: Ice Nucleation Activity of Graphene and Graphene Oxides, The Journal of Physical Chemistry C 122 (15), pp. 8182-8190, 2018. DOI: 10.1021/acs.jpcc.7b10675 Interactive comment on Atmos. Chem. Phys. Discuss., https://doi.org/10.5194/acp-2018-583, 2018.

References mentioned in our answers (which are not part of the originally submitted manuscript):

Emersic, C., Connolly, P. J., Boult, S., Campana, M., and Li, Z.: Investigating the discrepancy between wet-suspension- and dry dispersion-derived ice nucleation efficiency of mineral particles, Atmospheric Chemistry and Physics, 15, pp. 11 311–11 326, 2015.

Hiranuma, N., Brooks, S. D., Auvermann, B. W., and Littleton, R.: Using environmental scanning electron microscopy to determine the hygroscopic properties of agricultural aerosols, Atmospheric Environment, 42, pp. 1983–1994, 2008.

Gieré, R., Carleton, L. E., and Lumpkin, G. R.: Micro-and nanochemistry of fly ash from a coal-fired power plant, American Mineralogist, 88, 1853–1865, 2003.

Weinbruch, S., Ebert, M., Gorzawski, H., Dirsch, T., Berg, T., & Steinnes, E.: Characterisation of individual aerosol particles on moss surfaces: implications for source apportionment. Journal of Environmental Monitoring, 12(5), 1064-1071, 2010.